# SARS-CoV-2 vaccination elicits broad and potent antibody effector functions to variants of concern in vulnerable populations

Andrew P. Hederman[1], Harini Natarajan[2], Leo Heyndrickx[3], Kevin K. Ariën [3], Joshua A. Wiener [1], Peter F. Wright[4], Evan M. Bloch [5], Aaron A. R. Tobian[5], Andrew D. Redd[6,7], Joel N. Blankson[6], Amihai Rottenstreich[8], Gila Zarbiv[9], Dana Wolf[9], Tessa Goetghebuer [10,11], Arnaud Marchant[10] & Margaret E. Ackerman [1,2] ✉

SARS-CoV-2 variants have continuously emerged in the face of effective vaccines. Reduced neutralization against variants raises questions as to whether other antibody functions are similarly compromised, or if they might compensate for lost neutralization activity. Here, the breadth and potency of antibody recognition and effector function is surveyed following either infection or vaccination. Considering pregnant women as a model cohort with higher risk of severe illness and death, we observe similar binding and functional breadth for healthy and immunologically vulnerable populations, but considerably greater functional antibody breadth and potency across variants associated with vaccination. In contrast, greater antibody functional activity targeting the endemic coronavirus OC43 is noted among convalescent individuals, illustrating a dichotomy in recognition between close and distant human coronavirus strains associated with exposure history. This analysis of antibody functions suggests the differential potential for antibody effector functions to contribute to protecting vaccinated and convalescent subjects as novel variants continue to evolve.

Severe acute respiratory syndrome coronavirus-2 (SARS-CoV-2), the virus causing the COVID-19 pandemic, has continued to evolve despite the widespread use of multiple highly effective vaccines[1–3]. Notable SARS-CoV-2 variants have arisen and expanded across the globe, including Alpha (B.1.1.7), Beta (B.1.351), Gamma (P.1), Delta (B.1.617.2), and most recently, Omicron (B.1.1.529). While vaccination is a critical intervention in combatting the pandemic, many studies of both infection and vaccination have shown reduced neutralizing potency toward variants of concern (VOC)[4–14].

Fortunately, studies have also shown that both infection and vaccination lead to measurable levels of antibodies and T-cell responses against VOC[15–23]. Because binding antibodies can elicit potent antibody Fc domain-dependent effector functions that contribute to protection even in the absence of neutralization[24–27], they

[1]Thayer School of Engineering, Dartmouth College, Hanover, NH, USA. [2]Department of Immunology and Microbiology, Geisel School of Medicine at Dartmouth, Dartmouth College, Hanover, NH, USA. [3]Department of Biomedical Sciences, Institute of Tropical Medicine, Antwerp, Belgium. [4]Department of Pediatrics, Geisel School of Medicine at Dartmouth, Dartmouth-Hitchcock Medical Center, Lebanon, NH, USA. [5]Department of Pathology, Johns Hopkins School of Medicine, Baltimore, MD, USA. [6]Department of Medicine, Division of Infectious Diseases, Johns Hopkins School of Medicine, Baltimore, MD, USA. [7]Division of Intramural Research, National Institute of Allergy and Infectious Diseases, National Institutes of Health, Bethesda, MD, USA. [8]Department of Obstetrics and Gynecology, Hadassah-Hebrew University Medical Center, Jerusalem, Israel. [9]Clinical Virology Unit, Hadassah University Medical Center, Jerusalem, Israel. [10]Institute for Medical Immunology, Université libre de Bruxelles, Charleroi, Belgium. [11]Pediatric Department, CHU St Pierre, Brussels, Belgium. ✉e-mail: margaret.e.ackerman@dartmouth.edu

may help to compensate for compromised neutralization potency. These activities, including antibody-dependent cellular phagocytosis (ADCP), cellular cytotoxicity (ADCC), and complement deposition (ADCD), rely on interactions with soluble and cell-expressed antibody Fc receptors on diverse innate immune cells that can drive direct lysis of virions or infected cells, as well as trigger inflammatory cascades to amplify host defense[6,13,28–30]. Fc effector functions have correlated with protection in animal models of SARS-CoV-2 and in humans in other disease settings[27,31–34]. The potential clinical relevance of antibody-mediated Fc effector functions is suggested by observations that vaccines appear to remain highly effective in preventing serious disease despite reduced levels of neutralizing antibodies.

Importantly, studies have shown that Fc effector activities are elicited following vaccination and infection[25,28,35], and antibodies with the capacity to elicit multiple effector functions appear to recognize diverse epitopes on the spike (S) protein[25,36]. SARS-CoV-2 variants generally accrue most mutations in the receptor binding domain (RBD) and the N terminal domain (NTD), where the most common epitopes for neutralizing Abs are found[37–39]. The emergence of the Omicron variant, with far more mutations than other VOC, including in other regions of the spike protein, has led to reductions in neutralization potency against this variant[8,40,41]. These observations have raised interest in determining whether and which other antibody activities may help confer protection from severe disease from sequence distant strains and how well these responses are induced by infection or vaccination.

To this end, immunity resulting from vaccination and natural infection are known to exhibit a number of distinctions with respect to antibody responses[15,42,43]. Within groups, individuals also show considerable variability in response magnitudes and characteristics[44–46]. Lastly, populations at increased risk for severe disease, such as pregnant women[47] may also exhibit further distinctions in humoral response attributes[48]. How these distinct axes of variability associate with the magnitude and breadth of antibody effector functions across VOC is not known, but it has important implications for continued protection of diverse individuals and in the face of further viral diversification. The breadth of Fc effector functions across VOC in vulnerable and healthy subjects following natural infection or vaccination can provide new insight into the potential contributions of mechanisms beyond neutralization to protection from SARS-CoV-2 disease.

## Results

### Distinct Ig isotype responses induced by infection and vaccination across SARS-CoV-2 VOC, independent of immunological vulnerability

To study antibody Fc mediated effector functions across SARS-CoV-2 VOC, serum samples from individuals who either received two doses of an approved mRNA vaccine ($n = 87$), were previously infected when the Wuhan strain was predominant ($n = 57$) or were SARS-CoV-2 naïve ($n = 38$) (Supplemental Table 1) were first evaluated for the magnitude and characteristics of antibody responses to various CoV strains and subdomains (Supplemental Table 2). As a model of a uniquely vulnerable population, a subset of samples from vaccinated ($n = 50$) and convalescent ($n = 38$) individuals were collected from pregnant women who are at greater risk of hospitalization for covid-19[47]. To define similarities and differences in antibody profiles among seropositive individuals, dimensionality reduction was performed on biophysical antibody features using Uniform Manifold Approximation and Projection (UMAP)[49]. Subjects were distributed across the profile landscape into three distinct clusters, which were almost perfectly segregated by whether their responses were elicited by vaccination or infection (Fig. 1A). In contrast, major differences between pregnant and non-pregnant individuals were not observed, motivating combined analysis of subjects at different risk levels for subsequent

analysis. With the exception of time since most recent antigen exposure, which differed between vaccinated and convalescent individuals, relationships between cluster groups and other available clinical/demographic characteristics were not apparent.

IgM, IgA, and IgG responses to both the complete spike extracellular domain or the RBD from the ancestral Wuhan strain as well as Alpha, Beta, Gamma, Delta, and Omicron differed among convalescent and vaccinated subjects and naïve controls (Fig. 1B). Despite collection at a somewhat later timepoint following antigen exposure, IgM responses were elevated among individuals in the natural infection cohort. In contrast, vaccinated subjects had higher levels of IgG antibodies to both spike and RBD across diverse strains. Medians and ranges in IgA response magnitudes were generally similar between groups, though responses to RBD from some strains appeared bimodal among convalescent individuals with a dichotomy of high and low responders. Distinctions between the two clusters of convalescent subjects in the UMAP analysis could be explained by such bimodal distributions.

To begin to support the generalization of observations about relative responses across diverse VOC, correlations were calculated between Wuhan and other VOC for each antibody isotype in each subject group. As can be seen for Omicron, the most sequence-distinct variant tested (Fig. 1C), responses were generally more strongly correlated for IgG than IgA and for vaccinated subjects than convalescent subjects (Fig. 1D). For IgG responses, strong correlations were observed between variants, providing evidence that subjects with high levels of antibody binding to WT are also likely to have high levels of antibodies that bind to future variants. Likewise, these results suggest that individuals who generate lower levels of antibodies to contemporaneously circulating strains will have lower levels of antibodies that cross-react to future variants.

For a subset of subjects, serum neutralization of Wuhan and Omicron strains was evaluated (Fig. 1E). Whereas sera from both vaccinated and convalescent subjects neutralized the Wuhan strain to varying extents, Omicron neutralizing activity above the limit of detection was not observed in the majority of sera.

### The breadth of antibody responses varies by antigen exposure history and isotype

Because the breadth of the antibody binding responses across VOC may relate to infection risk for future variants, antibody binding breadth was assessed for each subject. Breadth-potency curves and breadth scores, defined as the geometric mean response across variants, were calculated for IgM, IgA, and IgG (Fig. 2A–C) for the panel of VOC across the full spike extracellular domain or only the RBD. Each isotype showed a distinct breadth profile. IgM breadth was greater in convalescent than vaccinated subjects (Fig. 2A). IgA breadth was similar (Fig. 2B), and IgG breadth was considerably greater among vaccinated individuals (Fig. 2C). Differences in breadth were somewhat more pronounced for RBD than for spike.

### Vaccination induces breadth in IgG subclasses and FcγR binding-propensity across SARS-CoV-2 VOC

The robust breadth of the overall IgG response led us to further explore potential differences in subclass and FcγR-binding capacity among antibodies to VOC (Supplemental Fig. 1), as both are known to mediate differences in antibody effector functions[34,50,51]. Breadth-potency curves and breadth scores, which represent composite measures of relative binding levels of antibodies in each serum sample across the panel of VOC, for each IgG subclass against the spike and RBD antigens were calculated (Fig. 3A). Each IgG subclass response was broader and more potent across VOC in seropositive subjects than in naïve controls, and breadth and potency was also universally greater in vaccinated than convalescent subjects. However, the relative magnitude of these differences varied among subclasses. Differences

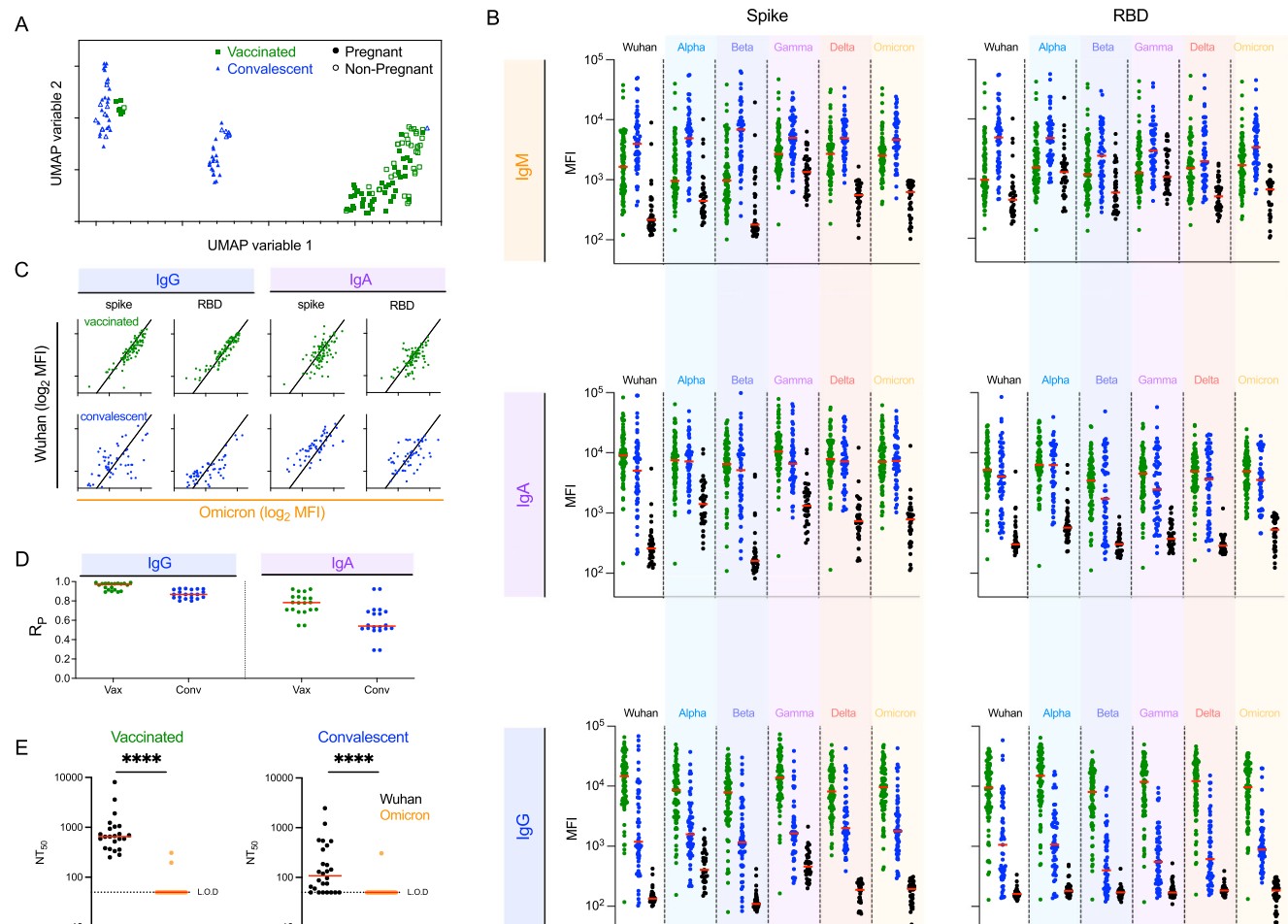

**Fig. 1 | IgM, IgA, and IgG antibody responses to VOCs following mRNA vaccination or natural infection. A** Coronavirus-specific antibody response features after dimensional reduction in pregnant ($n = 88$) (filled) and non-pregnant ($n = 56$) (open) individuals who were previously infected ($n = 57$) (blue triangles) or vaccinated ($n = 87$) (green squares). **B** Median Fluorescent Intensity (MFI) of IgM (top), IgA (center), and IgG (bottom) responses to spike (left) and RBD (right) of SARS-CoV-2 VOCs as defined by multiplex assay. Responses among SARS-CoV-2 naïve subjects ($n = 37$) are shown in black. Bar indicates the median response. **C** Representative scatterplots between IgG (left) and IgA (right) responses specific for Wuhan ($y$-axis) and Omicron ($x$-axis) in subjects following vaccination (top) or infection (bottom). The diagonal line indicates $x = y$. **D** Pearson correlation coefficient ($R_P$) for IgG (left) and IgA (right) responses across all pairs of variants in vaccinated (green) and infected (blue) subjects. **E** Neutralization titers ($NT_{50}$) were observed for serum samples from vaccinated ($n = 23$) (left) and convalescent ($n = 26$) (right) subjects against Wuhan (black) and Omicron (orange) strains. The limit of detection (LOD) is indicated by a dotted line. Statistical significance was defined by a two-sided Mann–Whitney test (****$p < 0.0001$). Bars indicate the median.

between seropositive subject groups were most pronounced for IgG1, IgG2, and IgG3, whereas IgG4 responses, though distinct from controls, were low in both groups of seropositive subjects. Compared to IgG1 and IgG3, IgG2 responses in convalescent subjects were more similar to naïve subjects than to those observed in vaccinated individuals. Collectively, these profiles show the greatest magnitude in antibody breadth for the most cytotoxic IgG subclasses (IgG1 and IgG3), intermediate breadth for moderately cytotoxic IgG2, and low breadth and potency for the relatively inert IgG4 subclass. Again, differences in the breadth among subclasses between vaccinated and convalescent subjects tended to be greater in RBD than the whole spike.

To further explore potential differences in the antiviral activity of SARS-CoV-2-specific antibody responses, their ability to bind to recombinant FcγR tetramers was assessed. Again breadth–potency curves and breadth scores were calculated (Fig. 3B) and were universally elevated among seropositive subjects relative to naïve controls and in vaccinated subjects relative to convalescent subjects. In general, the magnitude of differences in breadth and potency between vaccinated and convalescent subjects were greater in FcγR binding than in measures of individual IgG subclasses or total IgG. These

differences were again somewhat greater for RBD than the whole spike and exhibited differences of up to almost two orders of magnitude in median breadth score, suggesting that antibodies elicited by vaccination may be highly functional against SARS-CoV-2 variants.

## Greater breadth of antibody effector functions across SARS-CoV-2 VOCs

While the neutralization potency of antibodies raised from immunization or infection with ancestral strains is known to be reduced toward new VOC[52,53], whether similar losses in the extent of antibody effector functions are observed is less well studied[11,35,54]. For each subject, phagocytosis, ADCC, and complement deposition activities were assessed using full-length spike and RBD antigens for a panel of VOC across three different serum concentrations (Fig. 4). Whereas antibody effector functions observed in serum from naïve subjects at even the highest concentration were negligible (Supplemental Fig. 2), serum from both seropositive subject populations exhibited a diversity of antibody functions. With the exception of complement deposition against Wuhan strain whole spike protein, functional antibody responses were equal or greater in vaccinated than convalescent subjects for both Wuhan and diverse VOC. Among effector functions

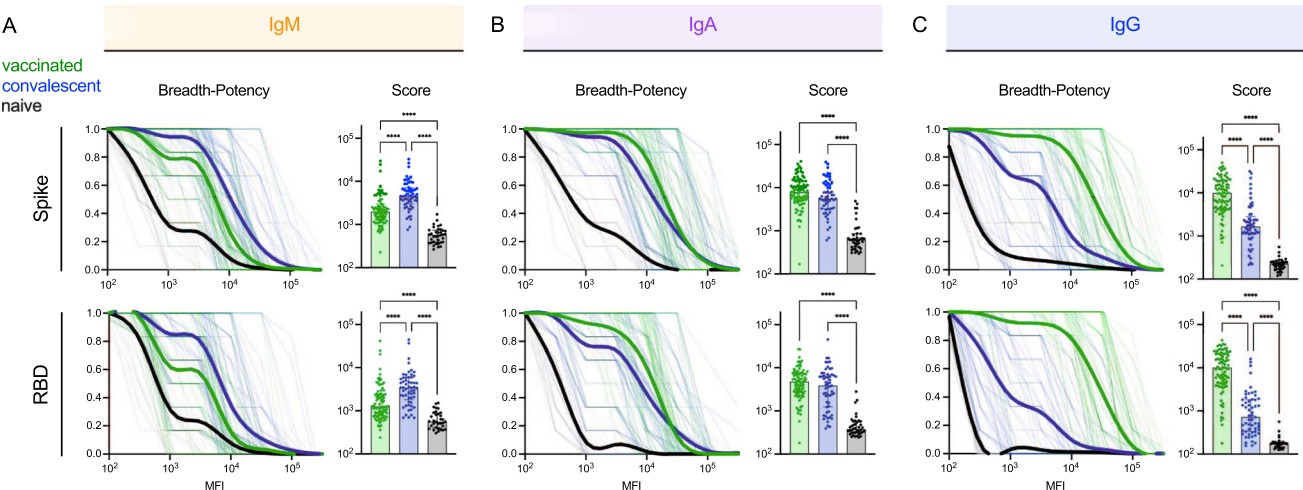

**Fig. 2 | Variable Ab response isotype breadth across VOC. A–C.** (Left) Breadth-potency curves represent the fraction of subjects with a response exceeding a given level for IgM (**A**), IgA (**B**), and IgG (**C**) antibody responses across the panel of VOC. Population means for naïve ($n = 37$) (black), vaccinated ($n = 87$) (green), and convalescent ($n = 57$) (blue) subjects is shown with a thick line, and individual subjects are illustrated in thin lines. (Right) Breadth scores for each subject. Bar indicates the median, and the whiskers indicate the interquartile range. Statistical significance was defined by ANOVA Kruskal–Wallis test with Dunn's correction and $\alpha = 0.05$ (****$p < 0.0001$). Responses to spike are shown at the top and to RBD at the bottom.

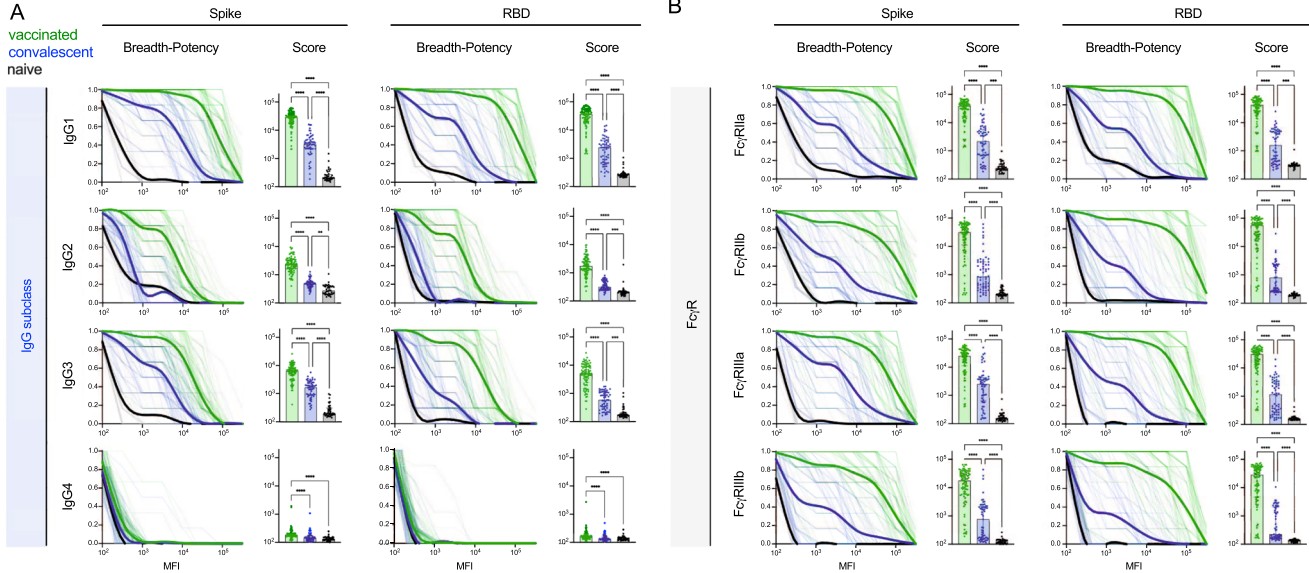

**Fig. 3 | Vaccinated subjects exhibit potentiated IgG subclass and FcγR-binding responses across VOC. A** (Left) Breadth–potency curves representing the fraction of subjects with a response exceeding a given level for IgG1–IgG4 antibody responses across the panel of VOC. Population mean for naïve ($n = 37$) (black), vaccinated ($n = 87$) (green), and convalescent ($n = 57$) (blue) subjects is shown with a thick line, and individual subjects are illustrated in thin lines. (Right) IgG subclass breadth scores for each subject. Bar indicates the median, and the whiskers indicate the interquartile range. **B** (Left) Breadth-potency curves representing the fraction of subjects with a response exceeding a given level for FcγRIIa, FcγRIIb, FcγRIIIa,

and FcγRIIIb-binding antibody responses. Population mean for naïve ($n = 37$) (black), vaccinated ($n = 87$) (green), and convalescent ($n = 57$) (blue) subjects is shown with a thick line, and individual subjects are illustrated in thin lines. (Right) IgG subclass breadth scores for each subject. Bar indicates the median, and the whiskers indicate the interquartile range. Statistical significance was defined by ANOVA Kruskal–Wallis test with Dunn's correction and $\alpha = 0.05$ (**$p < 0.01$, ***$p < 0.001$****$p < 0.0001$). Response to spike is shown at left and to RBD at right. Numerical values are presented as median fluorescent intensity (MFI).

tested, complement deposition, which often exhibits a steep dose-response profile, was the activity most strongly impacted by strain differences, for example, showing high activity in convalescent subjects only for Wuhan and alpha strains. This function was also the most sensitive to serum concentration and, perhaps as a result, showed greater differences in activity levels across VOC. Whereas phagocytosis and ADCC activity across VOC were reasonably well conserved at high and intermediate serum concentrations and often remained detectable at the lowest serum concentration in vaccinated subjects, decreases in activity were more pronounced in complement

deposition activity and were apparent at the intermediate concentration. Functional responses to RBD were somewhat more sensitive to decreasing serum concentration than were those to spike. This difference was most apparent in ADCC and complement deposition activities observed against beta and gamma strain RBD in vaccinated subjects. Despite lower levels of IgG1, IgG3, and C1q binding of spike/RBD-specific antibodies, convalescent subjects exhibited greater complement deposition compared with the vaccinated cohort against the Wuhan spike, an observation that may relate to contributions from IgM, which was elevated among convalescent individuals. Overall,

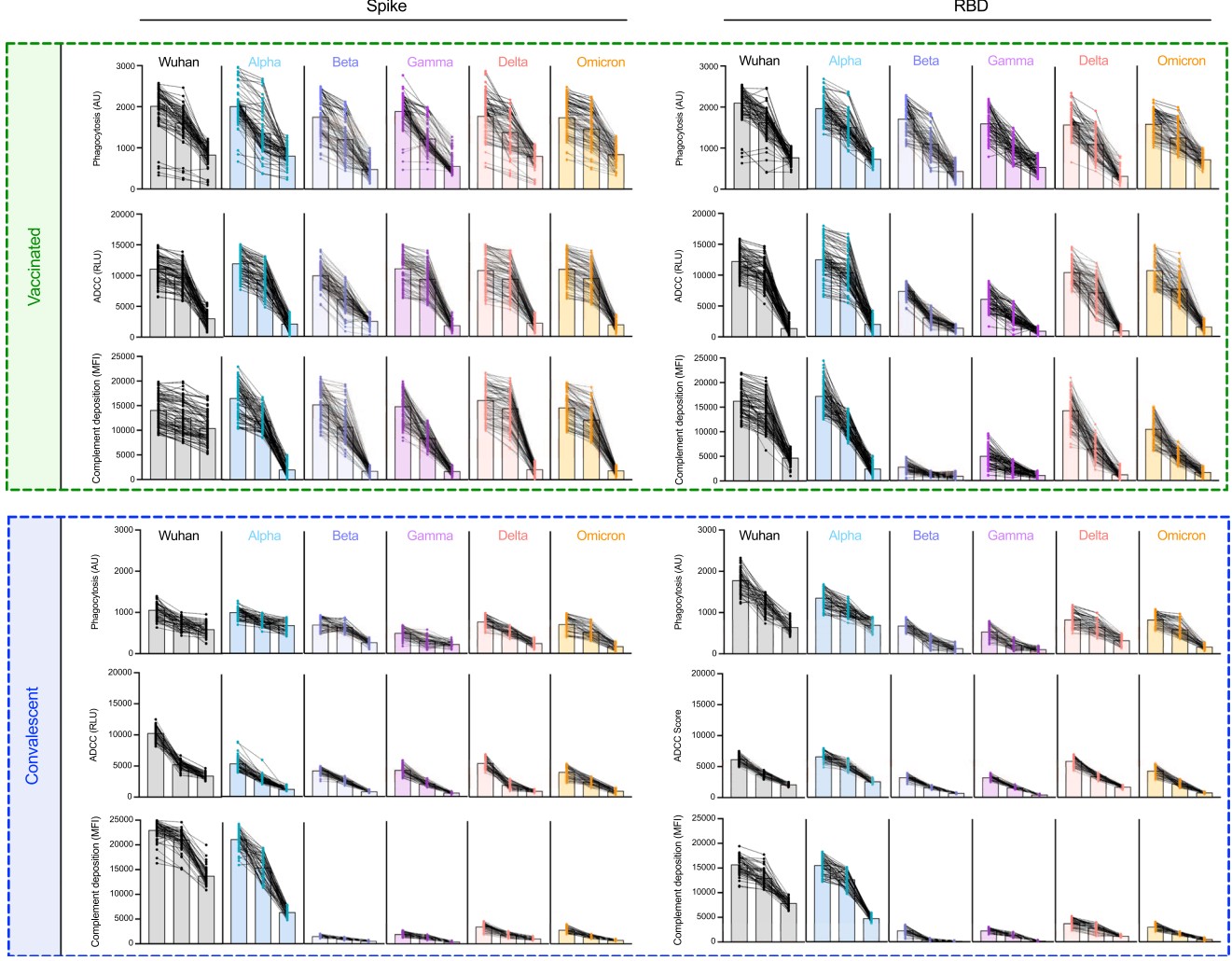

**Fig. 4 | mRNA vaccination results in the superior breadth of SARS-CoV-2-specific Ab effector function.** Ab effector function activities directed to spike (left) and RBD (right) in vaccinated (*n* = 87) (top) and convalescent (*n* = 57) (bottom) subjects. Phagocytosis, ADCC, and Complement deposition activities were assessed at each of three serum dilutions (1:50, 1:100, 1:250) for each indicated SARS-CoV-2 variant. Individual traces for each subject across dilutions are displayed. Bar indicates mean activity. Functional activity is reported in arbitrary units (AU), relative light units (RLU), and median fluorescent intensity (MFI).

breadth scores for each effector function showed the dramatically improved breath of functional antibody responses across SARS-CoV-19 variants among vaccinated as compared to convalescent subjects toward both spike and RBD (Fig. 5A), and suggest that viral variation has a greater effect on neutralization than on Fc-dependent effector functions.

**Functional dichotomy observed across more distant Coronaviruses**

While breadth of effector function across SARS-CoV-2 variants that have arisen during the pandemic is important as new VOC arise, we also wanted to explore whether Fc effector functions may be elicited towards more distant coronaviruses, including pathogenic coronaviruses, such as SARS-CoV-1 and MERS, which share 80% and 40% sequence identity to SARS-CoV-2[55,56], as well as against endemic coronaviruses including beta CoV OC43 and HKU1, and alpha CoV NL63, and 229E that historically account for 10–15% of respiratory infections in children and adults each year[57]. We observed a striking dichotomy between vaccinated and convalescent subjects in terms of phagocytosis, ADCC, and complement deposition activities across coronavirus strains. While robust SARS-CoV-1-specific effector function was observed for whole S and the S1 domain in vaccinated subjects, cross-reactive functional antibodies to these targets were not observed

among convalescent subjects (Fig. 5B). In contrast, robust activity toward whole S and the S2 domain of the endemic coronavirus OC43 was detected among convalescent subjects (Fig. 5C). Interestingly, the differences in antibody function were greater than might have been expected based on binding IgG antibody profiles observed across endemic CoV antigens (Supplemental Fig. 3), perhaps again reflecting contributions from other isotypes.

Some phagocytic activity was seen against HKU1 in convalescent subjects, although to a lesser extent than OC43. Vaccinated individuals showed some phagocytic activity toward these targets but no evidence of ADCC or complement deposition. Intriguingly, despite their lower S and S2-specific phagocytic activity, vaccinated subjects exhibited greater phagocytic activity directed to stabilized OC43 spike (S2P), suggesting that not only sequence but conformational state are key factors in defining functional antibody cross-reactivity profiles among even distantly related coronaviruses, as has recently been described for neutralization and binding activity[58,59]. Neither subject group exhibited functional responses to MERS S or its S1 domain or to other alphacoronaviruses 229E and NL63. In sum, both natural infection and vaccination appear to induce antibodies with better-maintained effector function breadth than neutralization breadth. However, the dichotomy in functional breadth between VOC, emergent, and endemic CoV observed between vaccinated and convalescent subjects,

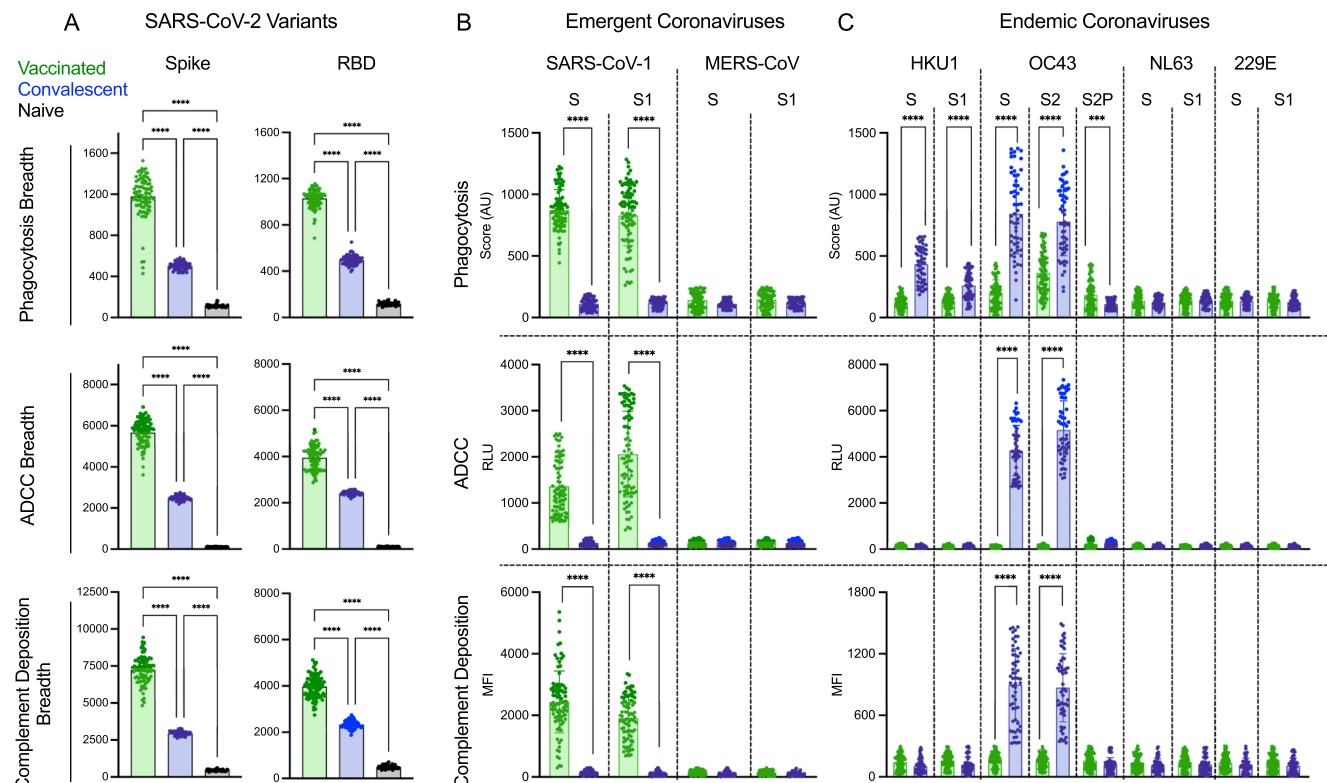

**Fig. 5 | Functional breadth across hCoV is imprinted by vaccination or infection history. A** Functional breadth in vaccinated ($n = 87$) (green), convalescent ($n = 57$) (blue), and naïve ($n = 37$) (black) subjects in phagocytosis (top), ADCC (center), and complement deposition (bottom) as defined by the geometric mean of each activity directed to spike (left) and RBD (right) across variants. Bar indicates mean, and whiskers indicate interquartile range. Statistical significance was defined by ANOVA Kruskal–Wallis test with Dunn's correction and $\alpha = 0.05$ (****$p < 0.0001$). **B**, **C**. Phagocytosis (top), ADCC (center), and complement deposition (bottom)

activities observed in vaccinated ($n = 87$) (green) and convalescent ($n = 57$) (blue) subjects across other emergent (**B**) and endemic (**C**) CoV spike antigens consisting of unstabilized (S) stabilized (S2P) and S1 or S2 subdomains of spike from indicated hCoV strains. Statistical differences were measured by the Mann–Whitney test (*$p < 0.05$, **$p < 0.01$, ***$p < 0.001$, ****$p < 0.0001$). Bars indicate the mean, and whiskers indicate the standard deviation. Functional activity is reported in arbitrary units (AU), relative light units (RLU), and median fluorescent intensity (MFI).

coupled with distinctions in antibody isotypes and preferences in recognition and functional targeting of different conformational states and subdomains (Supplemental Fig. 4), point to distinctions in responses that may relate to site and persistence of antigen exposure, conformation of antigen, or other factors that could be used to improve functional breadth in pursuit of "universal" CoV vaccines.

## Discussion

The continued emergence of SARS-CoV-2 variants has raised concern that vaccines based on the Wuhan strain will exhibit reduced efficacy over time in the face of viral evolution[14]. This fear is further exacerbated in vulnerable populations, among which vaccine effectiveness is already reduced[60,61]. Whereas many studies have shown that vaccination results in reduced neutralization of variants as compared to the Wuhan strain[30,53], vaccines remain protective from severe disease[62], suggesting that either a threshold effect exists or that contributions from other cellular and humoral mechanisms may be at play. This latter possibility is well-supported by studies providing evidence that Fc effector functions are more resistant to changes from VOC, pointing to a potential component of protection from severe disease[18,54].

The continually evolving viral landscape of SARS-CoV-2 strains shows the importance of needing to further understand how effector functions may provide protection from severe disease when vaccine neutralization is reduced. To this end, and relative to the more considerably reduced neutralization potencies expected, we report well-maintained antibody effector functions across diverse VOC. Broad effector function was observed for both vaccinated and convalescent subjects and for both whole spike and RBD, although the former was

somewhat broader than the latter. This data suggests that for VOC against which neutralization activity may be insufficient, Fc-mediated effector functions have the potential to compensate for decreased neutralization and contribute to protection.

This study shows that the breadth of antibody recognition across SARS-CoV-2 VOC varies by isotype and by antigen exposure history. Whereas vaccinated subjects showed considerably greater FcγR-binding IgG antibodies of diverse subclasses, convalescent subjects exhibited greater IgM breadth. IgA breadth was comparable between populations. The distinctions among isotypes may reflect differences in exposure route, duration, and costimulatory factors. As mucosal vaccines continue to advance toward the clinic, these possibilities can be tested. Our data also suggests that the differences observed in the breadth of recognition of VOC and emergent CoV as compared to endemic CoV between vaccination and infection likely at least partially relate to distinctions in antigenic conformations between native and proline-stabilized forms of spike[58,59,63]. Additionally, binding antibody breadth does not appear to simply scale with greater IgG response magnitude, as for each antigen tested, at least some convalescent subjects exhibited responses of similar magnitude to vaccinated subjects, yet no convalescent subject exhibited a similar breadth score. Instead of response magnitude, these differences in breadth across VOC may relate to differences at the level of B cell responses between vaccinees and convalescent individuals[6,64]. Considerable antibody breadth and potency across VOC were similarly observed for diverse antibody effector functions, including phagocytosis mediated by monocytes, a surrogate measure of ADCC, and complement deposition. Among these functions, complement deposition was generally

more sensitive to both antigenic variation and to serum concentration than the other activities tested; and among VOC, beta and gamma RBD variants showed a greater decrease in ADCC and complement deposition activity than more distant VOC, a pattern that was not easily explained by mutational loads or identities.

While the observation of greater breadth and potency induced by vaccination than natural infection could be extended to SARS-CoV-1, neither exposure induced functional antibodies to MERS, and depending on spike conformation, the opposite pattern was observed for the endemic CoV OC43. These observations establish an intriguing dichotomy: breadth across CoV-2 strains was greater among vaccine recipients, but breadth across more distant beta and alpha coronaviruses was greater among convalescent subjects. While the origin of this dichotomy cannot be defined from this study, the observation that functional breadth toward stabilized and unstabilized spikes differed between populations suggests that it may be related to the conformational state of the spike antigen to which the immune system was exposed. This data suggests differential antigenicity and immunogenicity of the stabilized spike protein used in mRNA vaccines and has important implications for efforts to develop pan-CoV vaccines.

We initially set out to characterize functional breadth in a vulnerable population but found only limited differences in the binding and functional profiles of antibodies in pregnant women as compared to healthy controls. While this observation has important implications for vulnerable populations, it is important to note that even in the context of similarly functional antibodies, the effector capacity of susceptible individuals may be compromised due to alterations in the cellular effectors and the availability or regulation of complement cascade factors. While the assays used herein have correlated with improved protection in vivo in various settings[65–69], only three activities were evaluated. Similarly, only five major VOCs were evaluated, and subjects were infected or vaccinated with the Wuhan strain. The breadth of effector functions induced by strains other than Wuhan or against future viral variants may exhibit different or similar degrees of conservation, as observed here. While some comparability testing was performed herein, other limitations include the use of surrogate measures of effector function reliant on recombinant antigens and cell lines as opposed to infected cells or virions and primary effectors. For example, the expression of regulatory factors expected to potentially impact complement deposition activity was not captured in the assay employed in this study.

Nonetheless, these observations of well-maintained functional antibody breadth following vaccination and infection, even among vulnerable individuals, have important implications. They suggest that antibodies elicited by either prior infection or vaccination with a given strain have the potential to restrict the replication of disease caused by future variants. Similarly, given evidence that these activities contribute to the efficacy of convalescent plasma[70] and monoclonal antibody therapy[24,27,71,72], our observations support the potential value of these interventions even in the face of continually diversifying viruses. Our observations of the enhanced breadth of IgG-dependent Fc effector functions provide additional evidence of beneficial aspects of vaccine-induced immunity as compared to natural infection. Future studies could define the impact hybrid immunity resulting from combinations of infection and vaccination or after a combination of distinct vaccine regimens and variant-specific boosters may have on functional breadth in healthy and immune-vulnerable populations. Overall, this work provides insights into how vaccines and prior natural infection may provide protection by antibody functional mechanisms. These observations promote vaccination as able to drive superior functional antibody breadth, and set expectations for cross-variant antibody effector function and may serve as a useful comparator in studies of candidate vaccines aimed at improving the breadth of protection and in evaluating population susceptibility as exposure histories and infecting virus continue to evolve.

## Methods

### Human subjects
This study complies with relevant human subject research regulations and was reviewed and approved by the Dartmouth College Committee for Protection of Human Subjects. Vaccinated subjects ($n = 87$) received two doses of either mRNA-1273 ($n = 2$) or BNT162b2 ($n = 85$) vaccines. Vaccinated subjects were pregnant women in Israel ($n = 50$) who were screened for lack of anti-N SARS-CoV-2 antibody responses or non-pregnant subjects from the United States ($n = 37$), among which four subjects had a prior history of SARS-CoV-2 infection. These subjects were included in the analysis as they were not clearly distinct from other vaccinated subjects in multivariate (UMAP) analysis. Convalescent subjects were pregnant women from Belgium ($n = 38$) or non-pregnant subjects from Dartmouth Hitchcock Medical Center in the United States ($n = 19$) with infection status defined by RT-PCR. Collection of these samples occurred when Wuhan was the dominant strain in circulation, but viruses were not typed. Naïve serum was obtained from a commercial source prior to the approval of vaccines and was screened for anti-N SARS-CoV-2 antibody responses to exclude donors with previous infection. Characteristics for each cohort are described in Supplemental Table 1. While pregnant subjects completed their vaccination series in the third trimester, and most convalescent subjects reported symptoms or tested positive in their third trimester, elapsed time since most recent SARS-CoV-2 antigen exposure differed between cohorts (Supplemental Fig. 5), which may impact some of the observations reported. Sex and/or gender were not considered in the study design or analysis. Subjects provided informed written consent, and studies were reviewed and approved by IRBs at individual collection sites and Dartmouth.

### Antigen and Fc receptor expression
Antigens were purchased from commercial sources or transiently expressed in Expi293 or HEK293 cells and purified via affinity chromatography following manufacturers' protocols (Supplemental Table 2). Fc receptors were expressed and purified as described previously[73].

### Fc array
Antigen-specific antibodies were characterized using the Fc array assay[74]. Briefly, antigens were covalently coupled to MagPlex microspheres (Luminex Corporation) using two-step carbodiimide chemistry. Experimental controls included pooled human polyclonal serum IgG (IVIG), S309, an antibody from a SARS-CoV patient that cross-reacts SARS-CoV and SARS-CoV-2, and VRC01, an HIV-specific antibody[75,76]. Serum dilutions used in experiments were based on experience from previous work and a small pilot experiment of test concentrations. Final dilutions used in assays varied from 1:250 to 1:5000 depending on the detection reagent (see reporting summary). Antigen-specific antibodies were detected by R-phycoerythrin-conjugated secondary reagents specific to human immunoglobulin isotypes and subclasses and by Fc receptor tetramers as described previously[77,78]. Median fluorescent intensity data was acquired on a FlexMap 3D array reader (Luminex Corporation). Samples were run in technical duplicates.

### Neutralization
SARS-CoV-2 neutralizing antibodies (nAb) were quantified[79] for a subset of sera samples (23 pregnant vaccinated and 26 pregnant convalescents, selected based on having the highest binding antibody levels from among subjects with sufficient serum volumes available). Briefly, serial dilutions of heat-inactivated serum (1/50 to 1/25,600 in EMEM supplemented with 2 mM L-glutamine, 100 U/ml–100 μg/mL of Penicillin–Streptomycin and 2% fetal bovine serum) were incubated for 1 hr at 37 °C and 7% $CO_2$ with 3×TCID$_{100}$ of Wuhan strain (2019-nCoV-Italy-INMI1, 008 V-03893) and Omicron strain BA.1 (B1.1.529,

VLD20211207). A volume of 100 µL of sample-virus mixture was added to 100 µL of Vero cells (18,000 cells/well) in a 96-well plate and incubated for 5 days at 37 °C and 7% $CO_2$. The cytopathic effect caused by viral growth was scored microscopically. The Reed-Muench method was used to calculate the nAb titer that reduced the number of infected cells by 50% ($NT_{50}$), which was used as a proxy for the nAb concentration in the sample. In accordance with WHO guidance, an internal reference standard composed of a pool of serum from naturally infected and vaccinated adults was included in each nAb assay run. This internal standard was calibrated against the Internationals Standard 21/234 (NIBSC), and $NT_{50}$ values were recalculated to IU/mL.

## Phagocytosis

Characterization of the phagocytic activity of serum antibodies was performed[80]. Briefly, 1 µM yellow-green fluorescent beads (Thermo Fisher, F8813) were covalently conjugated to spike or RBD antigens. Beads were then incubated with serum samples for 4 h with THP-1 cells (ATCC TIB-202) at 37 °C in 5% $CO_2$. Cells were fixed and analyzed by flow cytometry (Supplemental Fig. 6) using a MACSQuant Analyzer (Miltenyi Biotec). Scores were calculated as the percentage of cells that phagocytosed one or more fluorescent beads multiplied by the MFI of this population. S309 and VRC01 antibodies were included as positive and negative controls, respectively. Additional control wells with no added antibody were used to determine the level of antibody-independent phagocytosis. Serum samples were assayed at three different dilutions, which were determined by an initial pilot experiment to determine the optimal dilution series for measuring signal compared to the background. Concentrated pooled polyclonal serum IgG (Sigma Aldrich I4506) was used as a positive control for endemic CoV (Supplemental Fig. 7), and samples were run in three biological replicates.

## Reporter cell assay of antibody-dependent cellular cytotoxicity (ADCC)

A surrogate for antibody-mediated cellular cytotoxicity was measured using a CD16 reporter assay system[81]. Jurkat Lucia NFAT (Invivogen, jktl-nfat-cd16) cells were cultured according to manufacturer's instructions. Cultured cells express CD16 (FcγRIIIa), which, when engaged on the cell surface, leads to luciferase secretion from the cell. First, high-binding 96-well plates were coated overnight at 4 °C with 1 µg/mL of spike or RBD antigen. Following incubation, plates were washed (PBS + 0.1% Tween20) and blocked (PBS + 2.5% BSA) at room temperature (RT) for 1 h. Following plate washing, 100,000 cells per well and dilute serum samples were added to each well in cell culture media lacking antibiotics in a 200 µL volume. Following 24 h incubation, 25 µL of supernatant from each well was transferred into a white 96-well plate in which 75 µL of quantiluc substrate was immediately added. Following 10 min incubation, plates were read on a SpectraMax plate reader (Molecular Devices). VRC01 was used as a negative control; cell stimulation cocktail (Thermo Fischer Scientific, 00-4970-93) and ionomycin and S309 served as positive controls. Concentrated pooled polyclonal serum IgG (Sigma Aldrich I4506) was used as a positive control for endemic CoV (Supplemental Fig. 7), and samples were run in three biological replicates.

## ADCC against Spike-expressing cells

The antibody-dependent killing of Spike-expressing target cells was assessed[82,83] for a subset of samples (10 vaccinated, 8 convalescent, selected to cover the range of activity observed in the reporter cell assay), which were tested in order to evaluate the concordance between FcγRIIIa signaling determined in the reporter assay and cell killing (Supplemental Fig. 8A). Briefly, CEM.NKR (NIH.AIDS Reagent Program, ARP-4376) cells were transfected to express the SARS-CoV-2 spike (Sino Biological, VG40589-ACGLN) and serve as target cells. PBMCs were rested overnight and used as effector cells. Target cells were stained using cell tracker orange (Invitrogen, C3455), and effector cells were stained using cell tracker violet (Invitrogen, C100094). Target and effector cells were mixed at a ratio of 1:10 in RPMI media and incubated for 4 h at 37 °C in 5% $CO_2$. Serum from vaccinated and convalescent subjects was added at a 1:250 dilution. After incubation, cells were fixed and analyzed by flow cytometry (Supplemental Fig. 8B) on a MACSQuant Analyzer (Milltenyi) using FlowJo (version 10).

## Antibody-dependent complement deposition (ADCD)

Antibody-dependent complement deposition (ADCD) experiments were performed essentially as previously described[84]. Serum samples were first heat inactivated at 56 °C for 30 min. Samples were then incubated for 2 hr at RT with assay microspheres. The optimal dilutions for serum were determined from a small pilot experiment testing a range of dilutions. Human complement serum (Sigma, S1764) was diluted 1:100 in gel veronal buffer (Sigma-Aldrich, GVB++, G6514) and mixed with samples at RT with shaking for 1 h. After washing, samples were incubated with murine anti-C3b (Cedarlane #CL7636AP) at a 1:500 dilution at RT for 1 h, followed by staining with anti-mouse IgG1-PE secondary Ab (Southern Biotech #1070-09) at a 1:1000 dilution at RT for 30 min. A final wash was performed, and samples were resuspended into Luminex sheath fluid, and MFI was acquired on a FlexMap 3D reader. Assay controls with no antibody, an irrelevant VRC01 antibody, and a heat-inactivated complement were used as negative controls. S309 was used as a positive control. Concentrated pooled polyclonal serum IgG (Sigma Aldrich I4506) was used as a positive control for endemic CoV (Supplemental Fig. 7), and samples were run in three biological replicates.

## Statistics

Statistical analysis was performed in GraphPad Prism (version 9.7). Breadth–potency curves were defined as the proportion of antigen-specificities exhibiting a signal above a given intensity. Curves were generated using the LOWESS curve fit method in Prism for each respective subject group. Breadth scores were calculated by taking the geometric mean across antigen specificities for each subject. The sample size for each figure includes all subjects from their respective groups unless otherwise noted. Data graphed in figures is provided in Source Data File 1 along with statistical significance (ANOVA Kruskal–Wallis with Dunn's correction and $\alpha = 0.05$) of differences between groups.

## Reporting summary

Further information on research design is available in the Nature Portfolio Reporting Summary linked to this article.

# Data availability

Source data are provided in this paper. The antibody profile data generated and the statistical significance of group differences in this study are provided as Source Data File 1. Source data are provided in this paper.

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

## Acknowledgements

We would like to thank all participants who enrolled in this study and the study and laboratory staff who helped collect and process the samples. A number of antigen expression constructs were provided by Dr. Jason McLellan (UT Austin), and the positive control mAb S309 was provided by Dr. Jiwon Lee (Dartmouth). The following reagent was produced under HHSN272201400008C and obtained through BEI Resources, NIAID, NIH: Spike Glycoprotein Receptor Binding Domain (RBD) from SARS-Related Coronavirus 2, Wuhan-Hu-1 with C-Terminal Histidine Tag, Recombinant from Baculovirus, NR-52307. The following reagent was deposited by the Centers for Disease Control and Prevention and obtained through BEI Resources, NIAID, NIH: SARS-Related Coronavirus 2, Isolate USA-WA1/2020, NR-52281. This work was supported in part by the Division of Intramural Research, National Institute of Allergy and Infectious Diseases, as well as extramural support from the National Institute of Allergy and Infectious Diseases (R01AI120938, R01AI120938S1, and R01AI128779 to A.A.R.T.), National Heart Lung and Blood Institute (K23HL151826 to E.M.B.), National Cancer Institute (2 P30 CA 023108-41 to M.E.A.), National Institute of General Medical Sciences (P20-GM113132 BioMT Molecular Tools Core to M.E.A.) and National Institute of Allergy and Infectious Disease (1U19AI145825 to M.E.A.). A.M. is Research Director at the F.R.S., FNRS, Belgium.

## Author contributions

M.E.A. conceived the study, obtained funding, and supervised the research. A.P.H. and H.N. performed Fc array experiments. A.P.H. performed antibody functional assays. L.H. and K.K.A. performed neutralization experiments. P.F.W., E.W.B., A.A.R.T., A.D.R., J.N.B., A.R., G.Z., D.W., T.G., and A.M. contributed samples. A.P.H., J.A.W., and M.E.A. performed statistical analysis and wrote the original paper. All authors reviewed and edited the paper draft.

## Competing interests

The authors declare no competing interests.
