## [Peer Review File · Nature Communications]

SARS-CoV-2 vaccination elicits broad and potent antibody effector functions to variants of concern in vulnerable populationsREVIEWER COMMENTS

Reviewer #1 (Remarks to the Author):

This study by Hederman et al aims to characterize the Fc-mediated effector functions induced in the context of infection and vaccination across variants of concern across vulnerable and healthy subjects. This is a topic of great importance that could help explain how vaccination continues to provide some level of protection against severe disease following infection by SARS-CoV-2 variants, especially when neutralization is lost. Moreover, as the authors show the increased functional antibody breadth mediated by vaccine-induced antibodies, but not infection-induced antibodies, these data support vaccination strategies to combat emerging variants rather than relying on infection-induced antibodies alone. In addition, as the authors also analyze these responses in pregnant individuals and find that they are equivalent to non-pregnant individuals, these data add to the body of work that supports vaccination against SARS-CoV-2 in pregnancy.

The cohort analyzed is derived from the US and Europe, with subjects either vaccinated or recovered from SARS-CoV-2 infection, and from the data presented, vaccination/infection profiles are similar across these geographic locations. Binding of IgG, IgA, and IgM to 6 different SARS-CoV-2 variants (Wuhan, Alpha, Beta, Gamma, Delta, Omicron) were measured and the authors generate a “breadth score” to define the breadth of binding across variants, that encompasses the magnitude and coverage of responses against the 6 variants. They find that vaccination generates IgG with increased breadth compared to infection yet find that IgA breadth is equivalent, while IgM breadth is increased in infection. The observed increased breadth of IgG is maintained in the analysis of IgG subclasses and in the binding of FcγRs. Comprehensive analysis of the Fc-mediated effector functions (ADCP, ADCC, and ADCD) across all 6 variants is consistent with their IgG observations, and they find that vaccination increases functional breadth compared to infection, yet they also note some interesting differences in functional activity between variants. Notably, responses against beta and gamma RBD are reduced compared to the other variants in both vaccination and infection with ADCC and ADCD, but not ADCP. The authors also evaluate the functional responses against SARS-CoV-1, MERS-CoV, and the seasonal CoVs. Vaccine-induced responses are induced against SARS-CoV-1 but not MERS. Of note, the authors observe significantly elevated responses against OC43 and HKU1 in

infected individuals (against the S, S2), but the response against OC43 is diminished when the stabilized spike is used, and in that context increased responses in vaccinated individuals is observed. It is a perplexing piece of data to end with, but one that generates a lot of additional questions that may be explored in the future.

While this is not the first study to show that Fc-effector functions are retained in the face of global variants, it provides one of the most comprehensive analyses of Fc-effector across 6 major variants of SARS-CoV-2. The inclusion of the HCoVs in the analysis highlights that SARS-CoV-2 infection impacts immunity against related viruses, which may have implications for protection against these viruses. The study could benefit from some additional analyses and discussion to further explain some of their findings, but by and large, it is a nice, well-written, and well-considered study and I think it is of importance to the field.

Major points:

1. The methods mention that 4 vaccinated individuals had evidence of prior infection. Were these individuals included in the analysis? As hybrid immunity induced by infection followed by vaccination boosts levels of antibodies, which has been observed by many groups, it may be best to remove those 4 individuals from the analysis.
2. The UMAP in Figure 1A shows a beautiful split between vaccinated and convalescent populations. The convalescent population is further split into two populations. What are the antibody features driving that split? Looking at Figure 1B, it looks as if there might be two populations in some of the antibody measures. Alternatively, did these people experience any differences in disease severity? The middle UMAP population looks to be comprised mostly of pregnant people. Did the authors also include parameters such as location, disease severity, days post-PCR+/days post 2nd dose? Were the effector function responses presented in Figure 4/5 also included in the UMAP?
3. The multiplexed approach to measure antigen-specific antibody levels has the advantage of measuring multiple specificities in the same well. Are the authors concerned about competition of binding to variant antigens by the same antibodies, potentially leading to reduced responses against one antigen vs another? Do the antibodies have equivalent affinity across the variants, or is affinity skewed towards one variant over another?
4. As IgM is a potent inducer of ADCD, were the authors surprised at the drop-off in ADCD across variants in the convalescents, especially since IgM levels against the variants and

breadth score were elevated in convalescents? What does antibody binding to C1q look like or to MBL?

5. Figure 4 shows that ADCC/ADCD is reduced against Beta and Gamma RBD, but then increased back in Delta and Omicron. Are there any mutations that are common only to those variants that could explain these data?

6. I realize that the focus of this manuscript is on analysis of Fc-effector function and not neutralization, but I think it would be important to show that neutralization responses are indeed reduced across the variants (maybe at least comparing Wuhan to Omicron), which could further support the authors' conclusions that Fc-effector function remains intact across variants.

7. The data presented in Figure 5 demonstrates a strikingly elevated OC43-specific functional activity in convalescent individuals but not vaccinated – are antibodies against OC43 also elevated? The near complete loss of activity when assayed using the S2P suggests that the stabilized spike completely occludes the epitopes that may be recognized when presented as a monomer (is the S from Sino Biologics a monomer?) or just the S2 alone. Is this specific to OC43? I saw that the authors also include these different domains for SARS-CoV-2 (spike, S2P, the hexaprop, S1 and S2) in their Fc array. Is there any difference in functional activity between these as well? Alternatively, could there be a specific cross-reactive epitope that is only induced with SARS-CoV-2 infection that is occluded in the stabilized OC43 spike?

8. I'm a little surprised that there is minimal functional activity induced against HKU1 (for ADCC and ADCD) NL63 or 229E given the global presence of these viruses. Did the authors include a positive control for those particular antigens (a seropositive individual or IVIG, which should have some NL63 or 229E specific antibodies, for example) just to ensure that the effector assays are working well with these antigens?

9. The finding that IgA breadth is similar between infection and vaccination may have important implications for the role of different isotypes in mediating protection, especially in mucosal protection. The authors have not evaluated the potential role of IgA in mediating effector function across the variants. While THP-1 cells can express FcαR, so it is possible that IgA may be contributing to phagocytosis in these cells, perhaps neutrophil-mediated phagocytosis could be also explored for a subset of subjects (or binding to FcαR?)

Minor:

1. Figure 1A – the blue and green colors used in the figure may be difficult to parse for those with color-blindness or when viewed/printed in black and white. Perhaps the authors could use different shapes as well?
2. Table 1 – I didn't see the reference for Crowley et al. AI 2021 in the reference list, but I wanted to take a look at that paper to see if the group had previously reported the neutralization for this cohort.
3. Figure 2 – can the authors clarify whether the MFI on the X axis is the MFI of a single antigen or a composite of all antigens?
4. Figure 4 - The authors should include a line or indicate where the naïve individual responses fall to allow the reader to understand whether the baseline responses against a given variant.
5. Days post vaccine/days post PCR+/symptom onset – are these statistically equivalent?
6. Were the pregnant people vaccinated/infected prior to pregnancy or during pregnancy?

Reviewer #2 (Remarks to the Author):

Hederman et al. evaluated effector functionality of anti-SARS-CoV-2 RBD and spike antibodies elicited by vaccination (largely BNT162b2) and by infection, early in the pandemic when the Wuhan strain was predominant, in pregnant and non-pregnant subjects. While no difference between the pregnant and non-pregnant groups was found with respect to the antibody effector functionality, complement activation and Fc receptor activation, this is none-the-less an important observation.

Overall, the main message of the paper is that functional antibodies that reach forward with functional cross reactivity to the then future variants (beta to omicron) were more broadly elicited by vaccination than by natural infection. The innate functional profile of the convalescent antibodies was always inferior to the vaccine responses, excepting for complement activation against the spike/RBD of the ancestral and alpha variants. Cross reactivity of the Wuhan convalescent functional antibodies against subsequent strains was poor in contrast to the vaccine antibodies. While this conclusion was not unexpected it is a comprehensive dataset that is collected at the high standard of this group and is an important piece of the picture of protective humoral responses to SARS-CoV-2.

In short, the paper adds to the body of evidence to the benefit of vaccination for pregnant women and generally for functional protection against future strains. Such evidence is a valuable addition, but it is noted that there have been many serological studies of COVID19.

A weakness of these studies, noted by the authors in their discussion, is the disconnect between these surrogate assays of innate effector functions and how they relate to mechanisms of resistance in humans in vivo. How does spike or RBD linked onto a bead emulate the features of a virally infected cell for ADCC or phagocytosis or ADCD? While no in vitro assay can perfectly emulate in vivo mechanisms of protection, there is a need to bridge the gap between descriptive serology studies and the protective mechanisms in vivo. For example CD16 induced luciferase expression informs only an aspect of what is the possible complexity of NK killing of an infected target. While some validation of assays toward protective mechanisms may have been made in other fields (e.g. HIV), but extending some similar work into SARS-COV-2 would move beyond descriptive serology.

It was difficult to find what proteins were used in what assays. These details, as noted in discussion of Figure 5, are important and should be supplied.

The UMAP analysis, besides not distinguishing the pregnant and non-pregnant populations, segregated the convalescent subjects into two groups. There was little or no insight provided as to what this might mean mechanistically or for clinical severity. If developed this was a novel aspect of the study.

Point by point.

Line 80 and 81. Reduced levels of nAb but remaining protection against severe disease is interpreted as evidence for Fc effector functions being clinically relevant. But can't this be instead evidence of T cell immunity?

Line 112-114. ...antibody features using Uniform Manifold Approximation and Projection (UMAP) 50. Subjects were distributed across the profile landscape into three distinct clusters, which were almost perfectly segregated by whether their responses were elicited

by vaccination or infection (Figure 1A).

What are the antibody features that in the main drive the segregation of these population, is it mostly simple IgG titres for the vaccinated and convalescent populations, and the biomodal IgA titres to RBD that drives separation of the two convalescent groups?

Line 121f. Is the persistent high IgM in the convalescent cohort indicative of poor class switching? Is there further insight for this? Does the IgM avidity vary between the vaccine and convalescent cohort?

Line 130. ...responses were generally more strongly correlated, for RBD than spike,... (Figure 1D).

Figure 1D has no comparison of RBD and spike. Is data missing or is this referring to elsewhere, back to 1C?

Line 163-164. To further explore potential differences in the antiviral activity of SARS-CoV-2-specific antibody 164 responses, their ability to bind to recombinant FcγR tetramers was assessed.

What is the relationship between tetramer FcR binding and the effector functions measured, phagocytosis or ADCC? Is this just another way of measuring IgG1 and IgG3 subclass binding with a different secondary reagent? What FcγRIIIa allele is used? There is some IgG2 subclass component of the response. Does FcγRIIIa allele have any correlation to clinical features?

Line 176 For each subject, phagocytosis, ADCC, and 177 complement deposition activities were assessed..

The effector functional assays have some limitations and this is acknowledged by the authors (line 276f). For high throughput data surrogate assays are required but it would be very helpful to validate these approaches for SARS-CoV-2 with a smaller validation cohort and some more functional assays that more closely match in vivo protective mechanisms. The reported ADCC is NFAT driven luciferase induction downstream from FcγRIIIa in a Jurkat T cell line. Rather than ADCC this is FcγRIIIa activation. How does this assay relate to NK cell killing of a virally infected target cell or spike expressing target cell?

The phagocytosis assay uses THP-1 cells. Which FcγRs are the active players in this assay?

Are they the same as anticipated to be important on a monocyte or MDM? There is a lot of FcγRI on THP-1 cells. Does this dominate the assay? In this case why is FcγRI not measured in the FcγR binding assays? If so, is this assay only a 'functional measure' of IgG1 and IgG3 subclass titre?

Likewise for the C deposition assay. How does deposition on spike or RBD coated microspheres relate to what is observed on a virally infected target cell or spike expressing target cell?

While some of the sources of SARS-CoV-2 proteins are provided in the reagents table, some appear missing. There should be Wuhan, α, β, γ, δ, o (six) versions of the spike. Instead, there are four differently formatted versions/parts of spike, SARS-CoV-2 S, SARS-CoV-2 S1, SARS CoV-2 S2-P, SARS CoV-2 S-6P referred to in the table, but (other than for Figure 5) it is not explained in the text which are used in which experiment. (Are there $6 \times 4 = 24$ versions of spike used in these experiments). It would be helpful to know what the rationale is for using which ones in which experiments? As noted in the text, "conformational state are key factors," so this information needs to be added.

Also, it would be useful to the reader if there was description in the text what these are, or add extra descriptor columns to the reagent table. E.g. what amino acid #s are these RBDs? "Plasmid provided by Jason McLellan," the sequence should be reported.

Figure 4. Is the potent ADCD of the convalescent samples against the ancestral and alpha spike/RBD attributable to the high IgM titre in these samples? Can IgM depletion be performed (eg thermofisher anti-IgM) with a limited sample set or pool to test this?

Figure 5. The superior activities of the convalescent sera for OC43 over the vaccine sera is an interesting observation that is not extensively developed. Several ideas are proposed but not tested. Can the authors provide any further insight on this point.

We thank both reviewers for their time providing comments that we believe improve our revised manuscript. Point by point comments are answered here in red, and changes are tracked in the revised manuscript.

Reviewer #1 (Remarks to the Author):

This study by Hederman et al aims to characterize the Fc-mediated effector functions induced in the context of infection and vaccination across variants of concern across vulnerable and healthy subjects. This is a topic of great importance that could help explain how vaccination continues to provide some level of protection against severe disease following infection by SARS-CoV-2 variants, especially when neutralization is lost. Moreover, as the authors show the increased functional antibody breadth mediated by vaccine-induced antibodies, but not infection-induced antibodies, these data support vaccination strategies to combat emerging variants rather than relying on infection-induced antibodies alone. In addition, as the authors also analyze these responses in pregnant individuals and find that they are equivalent to non-pregnant individuals, these data add to the body of work that supports vaccination against SARS-CoV-2 in pregnancy.

The cohort analyzed is derived from the US and Europe, with subjects either vaccinated or recovered from SARS-CoV-2 infection, and from the data presented, vaccination/infection profiles are similar across these geographic locations. Binding of IgG, IgA, and IgM to 6 different SARS-CoV-2 variants (Wuhan, Alpha, Beta, Gamma, Delta, Omicron) were measured and the authors generate a “breadth score” to define the breadth of binding across variants, that encompasses the magnitude and coverage of responses against the 6 variants. They find that vaccination generates IgG with increased breadth compared to infection yet find that IgA breadth is equivalent, while IgM breadth is increased in infection. The observed increased breadth of IgG is maintained in the analysis of IgG subclasses and in the binding of FcγRs. Comprehensive analysis of the Fc-mediated effector functions (ADCP, ADCC, and ADCD) across all 6 variants is consistent with their IgG observations, and they find that vaccination increases functional breadth compared to infection, yet they also note some interesting differences in functional activity between variants. Notably, responses against beta and gamma RBD are reduced compared to the other variants in both vaccination and infection with ADCC and ADCD, but not ADCP. The authors also evaluate the functional responses against SARS-CoV-1, MERS-CoV, and the seasonal CoVs. Vaccine-induced responses are induced against SARS-CoV-1 but not MERS. Of note, the authors observe significantly elevated responses against OC43 and HKU1 in infected individuals (against the S, S2), but the response against OC43 is diminished when the stabilized spike is used, and in that context increased responses in vaccinated individuals is observed. It is a perplexing piece of data to end with, but one that generates a lot of additional questions that may be explored in the future.

While this is not the first study to show that Fc-effector functions are retained in the face of global variants, it provides one of the most comprehensive analyses of Fc-effector across 6 major variants of SARS-CoV-2. The inclusion of the HCoVs in the analysis highlights that SARS-CoV-2 infection impacts immunity against related viruses, which may have implications for protection against these viruses. The study could benefit from some additional analyses and discussion to further explain some of their findings, but by and large, it is a nice, well-written, and well-considered study and I think it is of importance to the field.

Major points:

1. The methods mention that 4 vaccinated individuals had evidence of prior infection. Were these individuals included in the analysis? As hybrid immunity induced by infection followed by vaccination

boosts levels of antibodies, which has been observed by many groups, it may be best to remove those 4 individuals from the analysis.

We did include the four individuals with prior infection. We agree there have been numerous publications highlighting differences in hybrid immunity. In the context of our study, these subjects were maintained on the basis of lacking distinction from other vaccinated subjects (e.g. in the UMAP analysis).

2. The UMAP in Figure 1A shows a beautiful split between vaccinated and convalescent populations. The convalescent population is further split into two populations. What are the antibody features driving that split? Looking at Figure 1B, it looks as if there might be two populations in some of the antibody measures. Alternatively, did these people experience any differences in disease severity? The middle UMAP population looks to be comprised mostly of pregnant people. Did the authors also include parameters such as location, disease severity, days post-PCR+/days post 2nd dose? Were the effector function responses presented in Figure 4/5 also included in the UMAP?

We also found the split among convalescent subjects to be an interesting result of the UMAP analysis, which relied on IgM, IgA, and IgG antibody binding to spike and RBD SARS-CoV-2 strains. The reviewer is correct to suggest that the bimodal distribution in antibody binding to some of the variant antigens, particularly for RBD, appears to be a major driver, as described in line 141-145. Relationships between UMAP clusters and available clinical/demographic characteristics such as location, disease severity, age, pregnancy, time since infection, etc., were not apparent.

3. The multiplexed approach to measure antigen-specific antibody levels has the advantage of measuring multiple specificities in the same well. Are the authors concerned about competition of binding to variant antigens by the same antibodies, potentially leading to reduced responses against one antigen vs another? Do the antibodies have equivalent affinity across the variants, or is affinity skewed towards one variant over another?

Experimentally, testing in the context of extremely high degrees (eg: 50-plex) of highly redundant (eg: identical) antigens can reduce signal in these assays. Fortunately, while we cannot exclude an effect of signal suppression based on multiplexing, all samples were tested under the same conditions and thus subject to the same degree of competition.

As to insights into antibody affinity, unfortunately, signal in this assay relates to both antibody prevalence and affinity, leaving us unable to isolate affinity to support comparison across different antigen-specificities.

4. As IgM is a potent inducer of ADCD, were the authors surprised at the drop-off in ADCD across variants in the convalescents, especially since IgM levels against the variants and breadth score were elevated in convalescents? What does antibody binding to C1q look like or to MBL?

Elevated ADCD among convalescents is consistent with their higher levels of SARS-CoV-2-specific IgM. Both the drop-off in activity across variants, and against different conformations of Wuhan Spike (**Supplemental Figure 4**) are of interest. In our experience, the ADCD assay exhibits a threshold-like effect, and we hypothesize that the levels of antibody against VOC may fall in the very steep part of curve.

Antigen-specific antibody binding to MBL was not evaluated in this study; antigen-specific antibody binding to C1q is presented below (**Response to Review Figure 1, part of Supplemental Figure 1**).

Response to Review Figure 1, part of Supplemental Figure 1. C1q responses to VOCs following vaccination or infection. Median fluorescent intensity of SARS-CoV-2 VOC spike- (A) and RBD- (B) specific antibody binding to C1q as defined by multiplex assay. Responses among SARS-CoV-2 naive subjects are shown in black. Bar indicates median response.

5. Figure 4 shows that ADCC/ADCD is reduced against Beta and Gamma RBD, but then increased back in Delta and Omicron. Are there any mutations that are common only to those variants that could explain these data?

Beta and Gamma RBD sequences share two mutations (E484K and N501Y) while also both having a mutation at position 417 albeit for different amino acids. The delta variant does not share these mutations/sites with Beta and Gamma. Omicron, however, does possess the K417N mutation also present in Beta as well as the N501Y mutation present in Beta and Gamma. Omicron also bears a mutation at position 484 like Beta and Gamma variants, though for a different amino acid. The only mutation shared between Omicron and Delta is T478K which is not present in Beta or Gamma variants (<https://doi.org/10.1016/j.cell.2022.01.001>). We agree with the reviewer that this is an interesting observation within our study, but find that these mutational patterns do not suggest an obvious explanation.

6. I realize that the focus of this manuscript is on analysis of Fc-effector function and not neutralization, but I think it would be important to show that neutralization responses are indeed reduced across the variants (maybe at least comparing Wuhan to Omicron), which could further support the authors' conclusions that Fc-effector function remains intact across variants.

The revised manuscript now includes neutralization titers for a subset of vaccinated (n=23) and convalescent (n=26) subjects against Wuhan and Omicron viruses (**new Figure 1E**, also below as **Response to Review Figure 2**). As anticipated, neutralization activity against the omicron variant was undetectable in nearly all subjects.

Response to Review Figure 2 and Revised Manuscript Figure 1E. Neutralization titers (NT_{50}) observed for vaccinated (left) and convalescent (right) subject serum samples against Wuhan (black) and Omicron (orange) strains. Limit of detection (LOD) is indicated in the dotted line. Red bar indicated median. Statistical significance was defined by Mann-Whitney test with ($****p < 0.0001$).

7. The data presented in Figure 5 demonstrates a strikingly elevated OC43-specific functional activity in convalescent individuals but not vaccinated – are antibodies against OC43 also elevated? The near complete loss of activity when assayed using the S2P suggests that the stabilized spike completely occludes the epitopes that may be recognized when presented as a monomer (is the S from Sino Biologicals a monomer?) or just the S2 alone. Is this specific to OC43? I saw that the authors also include these different domains for SARS-CoV-2 (spike, S2P, the hexaprotein, S1 and S2) in their Fc array. Is there any difference in functional activity between these as well? Alternatively, could there be a specific cross-reactive epitope that is only induced with SARS-CoV-2 infection that is occluded in the stabilized OC43 spike?

Yes, binding antibody responses to unstabilized forms of OC43, specifically the S2 domain, were also elevated in convalescent but not vaccinated subjects, as illustrated below (**Response to Review Figure 3, Supplemental Figure 3**). These results are consistent with more comprehensive prior reports (Crowley et al., eLife 2022, <https://doi.org/10.7554/eLife.75228>) investigating cross-reactivity. We also note that OC43 is not unique in this regard. For example, the other beta-CoV tested, HKU1, also showed evidence of elevated functional antibody responses (**Figure 5C, ADCP**) among convalescent but not vaccinated subjects.

We agree that the differing induction of endemic CoV-cross-reactive antibodies is intriguing. Our studies are consistent with differing antigenicity among spike domains and conformational states. Subdomains and distinct spike conformations were not available for all CoV strains (only those described in **Supplemental Table 2**). We include below the IgG binding profiles of all tested forms of the Wuhan strain (**Response to Review Figure 4A, Supplemental Figure 4A**).

We have also tested effector functions across different Wuhan domains/conformations (**Response to Review Figure 4B, now also included as Supplemental Figure 4B**).

Response to Review Figure 3 and Supplemental Figure 3. IgG responses to OC43 and HKU1 antigens. Median Fluorescent Intensity (MFI) levels of IgG specific for tested OC43 spike protein antigens (left) and HKU1 antigens (right) in serum among vaccinated (green), convalescent (blue), and naïve (black) subjects.

Response to Review Figure 4 and Supplemental Figure 4. IgM, IgA, and IgG antibody and effector function responses to SARS-CoV-2 Wuhan strain antigens. A. IgM, IgA, and IgG antibody responses of vaccinated (green), convalescent (blue), and naïve (black) against SARS-CoV-2 Wuhan strain spike domains. B. Functional activity of vaccinated (green), convalescent (blue), and naïve (black) serum samples observed against indicated SARS-CoV-2 Wuhan strain spike domains and conformations.

8. I'm a little surprised that there is minimal functional activity induced against HKU1 (for ADCC and ADCD) NL63 or 229E given the global presence of these viruses. Did the authors include a positive control for those particular antigens (a seropositive individual or IVIG, which should have some NL63 or 229E specific antibodies, for example) just to ensure that the effector assays are working well with these antigens?

We included either or both a high concentration of IVIG and a monoclonal Ab as positive controls, as presented below for endemic CoV (**Response to Review Figure 5 and Supplemental Figure 7**). Functional responses were low, perhaps unsurprising given the considerably lower binding antibody signal associated with these specificities as compared to SARS-CoV-2 antigens and the dilute serum used for analysis.

Response to Review Figure 5 and Supplemental Figure 7. Functional activity to endemic coronavirus antigens. Functional assay data for endemic CoV spike protein variants with high concentration serum-derived IgG (IVIG, gray) presented for comparison for dilute serum from vaccinated (green), convalescent (blue), and naïve (black) subjects.

9. The finding that IgA breadth is similar between infection and vaccination may have important implications for the role of different isotypes in mediating protection, especially in mucosal protection. The authors have not evaluated the potential role of IgA in mediating effector function across the variants. While THP-1 cells can express FcαR, so it is possible that IgA may be contributing to phagocytosis in these cells, perhaps neutrophil-mediated phagocytosis could be also explored for a subset of subjects (or binding to FcαR?)

We agree with the reviewer that IgA has an important role in the antibody response following vaccination and infection (Sterlin et al., *Sci Transl Med*, 2021, doi: 10.1126/scitranslmed.abd2223). Previous work from our group has explored ADNP activity (Butler et al., *Front Immunol*, 2021, doi: 10.3389/fimmu.2020.618685), which was observed to correlate with IgA and FcαR binding levels in

serum. Given the number of antigen-specificities tested, we did not consider assays that required fresh primary effector cells. However, in the revised manuscript we now report Fc α R binding (**Response to Review Figure 6 and Supplemental Figure 1**).

Response to Review Figure 6 and Supplemental Figure 1. Fc α R responses to VOCs following vaccination or infection. Median fluorescent intensity of SARS-CoV-2 VOC spike- (left) and RBD- (right) specific antibody binding to Fc α R as defined by multiplex assay. Responses among SARS-CoV-2 naïve subjects are shown in black. Bar indicates median response.

Minor:

1. Figure 1A – the blue and green colors used in the figure may be difficult to parse for those with color-blindness or when viewed/printed in black and white. Perhaps the authors could use different shapes as well?

We have updated the UMAP plot with different shapes as suggested.

2. Table 1 – I didn't see the reference for Crowley et al. AI 2021 in the reference list, but I wanted to take a look at that paper to see if the group had previously reported the neutralization for this cohort.

Neutralization data for a subset of samples in this study are now included as **Figure 1E**.

3. Figure 2 – can the authors clarify whether the MFI on the X axis is the MFI of a single antigen or a composite of all antigens?

In **Figure 2**, the X axis is a composite, which is now clarified in the legend.

4. Figure 4 - The authors should include a line or indicate where the naïve individual responses fall to allow the reader to understand whether the baseline responses against a given variant.

See **Response to Review Figures 3 and 4**. Additionally we have included **Response to Review Figure 7 and Supplemental Figure 2** to show naïve functional response to SARS-CoV-2 variants.

Response to Review Figure 7 and Supplemental Figure 2. Naïve antibody functional responses to SARS-CoV-2 VOC. ADCP, ADCC, and ADCD responses from naïve subjects shown in black for full length spike (left) and RBD (right) SARS-CoV-2 VOC antigens. Responses for naïve and vaccinated subjects are shown for the 1:50 dilution.

5. Days post vaccine/days post PCR+/symptom onset – are these statistically equivalent?

These are not statistically equivalent. We include this result as **Response to Review Figure 8 and Supplemental Figure 5.**

Response to Review Figure 8 and Supplemental Figure 5. Days post second vaccine dose and post symptom onset. Days post second vaccine dose for vaccinated subjects (green) and days post SARS-CoV-2 positive PCR result for convalescent subjects (blue). Statistical significance was defined by Mann-Whitney test with (**** $p < 0.0001$).

6. Were the pregnant people vaccinated/infected prior to pregnancy or during pregnancy?

Pregnant individuals were vaccinated during the third trimester of pregnancy (lines 348-350).

Reviewer #2 (Remarks to the Author):

Hederman et al. evaluated effector functionality of anti-SARS-CoV-2 RBD and spike antibodies elicited by vaccination (largely BNT162b2) and by infection, early in the pandemic when the Wuhan strain was predominant, in pregnant and non-pregnant subjects. While no difference between the pregnant and non-pregnant groups was found with respect to the antibody effector functionality, complement activation and Fc receptor activation, this is none-the-less an important observation.

Overall, the main message of the paper is that functional antibodies that reach forward with functional cross reactivity to the then future variants (beta to omicron) were more broadly elicited by vaccination than by natural infection. The innate functional profile of the convalescent antibodies was always inferior to the vaccine responses, excepting for complement activation against the spike/RBD of the ancestral and alpha variants. Cross reactivity of the Wuhan convalescent functional antibodies against subsequent strains was poor in contrast to the vaccine antibodies. While this conclusion was not unexpected it is a comprehensive dataset that is collected at the high standard of this group and is an important piece of the picture of protective humoral responses to SARS-CoV-2.

In short, the paper adds to the body of evidence to the benefit of vaccination for pregnant women and generally for functional protection against future strains. Such evidence is a valuable addition, but it is noted that there have been many serological studies of COVID19.

A weakness of these studies, noted by the authors in their discussion, is the disconnect between these surrogate assays of innate effector functions and how they relate to mechanisms of resistance in humans in vivo. How does spike or RBD linked onto a bead emulate the features of a virally infected cell for ADCC or phagocytosis or ADCD? While no in vitro assay can perfectly emulate in vivo mechanisms of protection, there is a need to bridge the gap between descriptive serology studies and the protective mechanisms in vivo. For example CD16 induced luciferase expression informs only an aspect of what is the possible complexity of NK killing of an infected target. While some validation of assays toward protective mechanisms may have been made in other fields (e.g. HIV), but extending some similar work into SARS-COV-2 would move beyond descriptive serology.

It was difficult to find what proteins were used in what assays. These details, as noted in discussion of Figure 5, are important and should be supplied.

We have clarified which proteins are used in each experiment within the text and figure legends.

The UMAP analysis, besides not distinguishing the pregnant and non-pregnant populations, segregated the convalescent subjects into two groups. There was little or no insight provided as to what this might mean mechanistically or for clinical severity. If developed this was a novel aspect of the study.

Please see our response to a similar comment made by Reviewer 1.

Point by point.

Line 80 and 81. Reduced levels of nAb but remaining protection against severe disease is interpreted as evidence for Fc effector functions being clinically relevant. But can't this be instead evidence of T cell immunity?

The revised manuscript now suggests additional factors, including cellular immunity, may contribute to protection (line 260-263).

Line 112-114. ...antibody features using Uniform Manifold Approximation and Projection (UMAP) 50. Subjects were distributed across the profile landscape into three distinct clusters, which were almost perfectly segregated by whether their responses were elicited by vaccination or infection (Figure 1A).

What are the antibody features that in the main drive the segregation of these population, is it mostly simple IgG titres for the vaccinated and convalescent populations, and the bimodal IgA titres to RBD that drives separation of the two convalescent groups?

Many features demonstrate relationships to UMAP groups, and bimodal titers do appear to play a role in the separation of two convalescent groups, as noted more fully above in our response to Reviewer 1, and in the manuscript text (line 141-146).

Line 121f. Is the persistent high IgM in the convalescent cohort indicative of poor class switching? Is there further insight for this? Does the IgM avidity vary between the vaccine and convalescent cohort?

This is an interesting question! Unfortunately, however, IgM avidity data is not available. We hypothesize that high levels of IgM may relate to route and conditions of exposure, but we cannot exclude a role of poor class switching.

Line 130. ...responses were generally more strongly correlated, for RBD than spike,... (Figure 1D).

Figure 1D has no comparison of RBD and spike. Is data missing or is this referring to elsewhere, back to 1C?

This error has been corrected.

Line 163-164. To further explore potential differences in the antiviral activity of SARS-CoV-2-specific antibody 164 responses, their ability to bind to recombinant FcγR tetramers was assessed.

What is the relationship between tetramer FcR binding and the effector functions measured, phagocytosis or ADCC? Is this just another way of measuring IgG1 and IgG3 subclass binding with a different secondary reagent? What FcγRIIIa allele is used? There is some IgG2 subclass component of the response. Does FcγRIIIa allele have any correlation to clinical features?

FcR multimer binding is often well correlated to effector functions and to the levels of activating IgG subclasses observed in SARS-CoV-2 (Lee et al., Cell Rep Med, 2021, doi: 10.1016/j.xcrm.2021.100296; Natarajan et al., BMC Immunol, 2021 <https://doi.org/10.1186/s12865-022-00480-w>; Butler et al., Front Immunol, 2021, doi: 10.3389/fimmu.2020.618685), as well as in a variety of other settings.

We note that while often correlated to IgG subclass profiles, FcR binding is impacted by additional factors such as allotype (Richardson et al., PLoS Path, 2019, <https://doi.org/10.1371/journal.ppat.1008064>) and IgG Fc glycosylation (Boesch et al., mAbs, 2014, <https://doi.org/10.4161/mabs.28808>).

The R131 variant of FcγRIIIa was used in this study (clarified in **Supplemental Table 2**).

As what feels like it might be the last bit of antibody biology to be evaluated in the context of SARS-CoV-2... we are unaware of studies in which host FcγRIIIa allotype has been studied in relation to disease severity or antibody responses in the context of SARS-CoV-2; subject genotype information is not available in this study.

Line 176 For each subject, phagocytosis, ADCC, and 177 complement deposition activities were assessed.

The effector functional assays have some limitations and this is acknowledged by the authors (line 276f). For high throughput data surrogate assays are required but it would be very helpful to validate these approaches for SARS-CoV-2 with a smaller validation cohort and some more functional assays that more closely match in vivo protective mechanisms.

The reported ADCC is NFAT driven luciferase induction downstream from FcγRIIIa in a Jurkat T cell line. Rather than ADCC this is FcγRIIIa activation. How does this assay relate to NK cell killing of a virally infected target cell or spike expressing target cell?

Below we demonstrate concordance between an NK killing assay and the FcγRIIIa reporter cell line assay for a subset of samples selected on the basis of representing a range of activity levels (**Response to Review Figure 9 and Supplemental Figure 8A**). These results are consistent with similar analysis conducted in Lee et al., *Cell Rep Med*, 2021 <https://doi.org/10.1016/j.xcrm.2021.100296>.

Response to Review Figure 9 and Supplemental Figure 8A. ADCC using an NK cell killing assay.

A. For a subset of vaccinated (green) and convalescent (blue) serum samples and NK cell killing assay was developed to determine the correlation between ADCC with a Jurkat reporter cell line and NK cell killing. R-squared was determined using a simple linear regression model. B. Sample gating strategy for NK cell killing assay.

The phagocytosis assay uses THP-1 cells. Which FcγRs are the active players in this assay? Are they the same as anticipated to be important on a monocyte or MDM? There is a lot of FcγRI on THP-1 cells. Does this dominate the assay? In this case why is FcγRI not measured in the FcγR binding assays? If so, is this assay only a 'functional measure' of IgG1 and IgG3 subclass titre?

FcγRII is the primary receptor (highest expression and greatest impact on bead uptake following blocking) associated with phagocytosis in the THP-1 assay (Ackerman et al., J Imm Methods, 2011, doi:10.1016/j.jim.2010.12.016). Phagocytosis induced by this receptor is strongly impacted by IgG subclass. We present antigen-specific antibody binding to FcγR1 below (**Response to Review Figure 10 and part of Supplemental Figure 1**).

Response to Review Figure 10 and part of Supplemental Figure 1. Antigen-specific antibody binding to FcγRI across spike and RBD proteins of Wuhan and VOC in serum from vaccinated (green), convalescent (blue), and naïve (black) subjects.

Likewise for the C deposition assay. How does deposition on spike or RBD coated microspheres relate to what is observed on a virally infected target cell or spike expressing target cell?

It is presently unclear how representative results from the bead-based complement deposition assay are to free virions, infected cells, or spike-expressing cells. While evidence from other settings suggests these activities can exhibit good concordance (Goldberg et al., mBio, 2021 <https://doi.org/10.1128/mBio.01743-21>), there are many complement regulatory factors that are not represented in the bead assay. This limitation is now better described (lines 315-317).

While some of the sources of SARS-CoV-2 proteins are provided in the reagents table, some appear missing. There should be Wuhan, α, β, γ, δ, o (six) versions of the spike. Instead, there are four differently formatted versions/parts of spike, SARS-CoV-2 S, SARS-CoV-2 S1, SARS CoV-2 S2-P, SARS CoV-2 S-6P referred to in the table, but (other than for Figure 5) it is not explained in the text which are used in which experiment. (Are there 6 x 4 = 24 versions of spike used in these experiments). It would be helpful to know what the rationale is for using which ones in which experiments? As noted in the text, “conformational state are key factors,” so this information needs to be added.

Supplemental Table 2 is complete. Stabilized and subdomain-level antigens were not tested for all variants or all epidemic/endemic CoV. The manuscript has been updated to specify which form of SARS-CoV-2 spike is reported in each figure.

Also, it would be useful to the reader if there was description in the text what these are, or add extra descriptor columns to the reagent table. E.g. what amino acid #s are these RBDs?

“Plasmid provided by Jason McLellan,” the sequence should be reported.

Additional detail and references are provided in Supplemental Table 2.

Figure 4. Is the potent ADCD of the convalescent samples against the ancestral and alpha spike/RBD attributable to the high IgM titre in these samples? Can IgM depletion be performed (eg thermofisher anti-IgM) with a limited sample set or pool to test this?

Our data is consistent with contributions of IgM to ADCD activity. Unfortunately, insufficient sample from these subjects was available for depletion experiments. This suggested experiment would be of great interest, however.

Figure 5. The superior activities of the convalescent sera for OC43 over the vaccine sera is an interesting observation that is not extensively developed. Several ideas are proposed but not tested. Can the authors provide any further insight on this point.

The revised manuscript now more fully discusses these observations (lines 243-247), which are consistent with those reported and more extensively developed in Crowley et al., eLife 2022, <https://doi.org/10.7554/eLife.75228>.

REVIEWER COMMENTS

Reviewer #2 (Remarks to the Author):

The revised manuscript by Hederman et al. has addressed my questions raised to the original submission.

New data includes correlating ADCC and the Jurkat reporter , neutralisation of omicron and Fc α RI binding enhances the submission.

Altogether it is a comprehensive and important analysis showing superior antibody effector functions induced by vaccination over that from infection by SARS-CoV-2. The likely significance of these effector functions is highlighted by the ineffectual neutralization of recent VOCs.

Minor points,

line 156 "Figure 1F" should be 1E

The symbols in the Y axis titles of supplementary Figure 1 are not rendered correctly.

Reviewer 1 in black text
Author response in red text.

Reviewer 1 provides a highly favourable assessment of this paper. The authors have provided new data sets in response to many of Reviewer 1's questions. Only some small changes to the manuscript text are suggested. Some clarification of a couple of the author's responses could be helpful, but in my opinion does not require another round of review. Statistical analysis of the supplementary figures should be provided.

Questions are in major and minor categories.

Major

1. The methods mention that 4 vaccinated individuals had evidence of prior infection. Were these individuals included in the analysis? As hybrid immunity induced by infection followed by vaccination boosts levels of antibodies, which has been observed by many groups, it may be best to remove those 4 individuals from the analysis.

We did include the four individuals with prior infection. We agree there have been numerous publications highlighting differences in hybrid immunity. In the context of our study, these subjects were maintained on the basis of lacking distinction from other vaccinated subjects (e.g. in the UMAP analysis).

Line 123f.

It is not made clear in the text that 4 individuals having hybrid immunity are included in the vaccine group. As reviewer 1 asks for clarification on this point it would be helpful for the text to make this clear and that their inclusion or exclusion did not affect the outcomes.

2. The UMAP in Figure 1A shows a beautiful split between vaccinated and convalescent populations. The convalescent population is further split into two populations. What are the antibody features driving that split? Looking at Figure 1B, it looks as if there might be two populations in some of the antibody measures. Alternatively, did these people experience any differences in disease severity? The middle UMAP population looks to be comprised mostly of pregnant people. Did the authors also include parameters such as location, disease severity, days post-PCR+/days post 2nd dose? Were the effector function responses presented in Figure 4/5 also included in the UMAP?

We also found the split among convalescent subjects to be an interesting result of the UMAP analysis, which relied on IgM, IgA, and IgG antibody binding to spike and RBD SARS-CoV-2 strains. The reviewer is correct to suggest that the bimodal distribution in antibody binding to some of the variant antigens, particularly for RBD, appears to be a major driver, as described in line 141-145. Relationships between UMAP clusters and available clinical/demographic characteristics such as location, disease severity, age, pregnancy, time since infection, etc., were not apparent.

Can line 145 be extended to also indicate other associations were not apparent?

3. The multiplexed approach to measure antigen-specific antibody levels has the advantage of measuring multiple specificities in the same well. Are the authors concerned about competition of binding to variant antigens by the same antibodies, potentially leading to reduced responses against one antigen vs another? Do the antibodies have equivalent affinity across the variants, or is affinity skewed towards one variant over another?

Experimentally, testing in the context of extremely high degrees (eg: 50-plex) of highly redundant (eg: identical) antigens can reduce signal in these assays. Fortunately, while we cannot exclude an effect of signal suppression based on multiplexing, all samples were tested under the same conditions and thus subject to the same degree of competition.

As to insights into antibody affinity, unfortunately, signal in this assay relates to both antibody prevalence and affinity, leaving us unable to isolate affinity to support comparison across different antigen-specificities.

Reviewer 1 has asked an interesting question on multiplexing methodology to which the authors have responded appropriately, but this should not affect the data presented or the text.

4. As IgM is a potent inducer of ADCD, were the authors surprised at the drop-off in ADCD across variants in the convalescents, especially since IgM levels against the variants and breadth score were elevated in convalescents? What does antibody binding to C1q look like or to MBL?

Elevated ADCD among convalescents is consistent with their higher levels of SARS-CoV-2-specific IgM. Both the drop-off in activity across variants, and against different conformations of Wuhan Spike (Supplemental Figure 4) are of interest. In our experience, the ADCD assay exhibits a threshold-like effect, and we hypothesize that the levels of antibody against VOC may fall in the very steep part of curve.

Antigen-specific antibody binding to MBL was not evaluated in this study; antigen-specific antibody binding to C1q is presented below (Response to Review Figure 1, part of Supplemental Figure 1).

C1q responses to VOCs following vaccination or infection. Median fluorescent intensity of SARS-CoV-2 VOC spike- (A) and RBD- (B) specific antibody binding to C1q as defined by multiplex assay. Responses among SARS-CoV-2 naïve subjects are shown in black. Bar indicates median response.

The authors have responded with a comprehensive data set of C1q binding for the cohorts which adds to an already comprehensive study. At present there is not my knowledge a way of dissecting IgG and IgM contribution to ADCD or C1q binding, although some specific inhibitors do exist- which might be used. The fall in ADCD in the convalescents is likely accounted for by responses being steep with respect to opsonising concentration, but it would be helpful to explain this in the text for the reader.

Further to this aspect pointed out by reviewer 1, it is very interesting that the higher IgG1/3 binding to Wuhan spike/RBD by the vaccinated group over the convalescents is matched by higher C1q binding (Supplemental Figure 1), however this higher C1q binding does not result in higher ADCD, since the convalescent cohort has clearly higher ADCD than the vaccinated (Supplemental Figure 4). I naively expected trends in C1q binding to result in matching trends in ADCD. I think this is an aspect that other readers would not anticipate, and so the authors could consider if it is useful to point out in the text. I.e. vaccinated group has highest IgG binding to Wuhan and highest C1q binding, but convalescent ADCD is superior.

5. Figure 4 shows that ADCC/ADCD is reduced against Beta and Gamma RBD, but then increased back in Delta and Omicron. Are there any mutations that are common only to those variants that could explain these data?

Beta and Gamma RBD sequences share two mutations (E484K and N501Y) while also both having a mutation at position 417 albeit for different amino acids. The delta variant does not share these mutations/sites with Beta and Gamma. Omicron, however, does possess the K417N mutation also present in Beta as well as the N501Y mutation present in Beta and Gamma. Omicron also bears a mutation at position 484 like Beta and Gamma variants, though for a different amino acid. The only mutation shared between Omicron and Delta is T478K which is not present in Beta or Gamma variants (<https://doi.org/10.1016/j.cell.2022.01.001>). We agree with the reviewer that this is an interesting observation within our study, but find that these mutational patterns do not suggest an obvious explanation.

This feature of the data is pointed to in the text of the results in line 217f. It is perhaps helpful to the reader to similarly extend line 292 in the discussion to re-point the reader to this interesting diminution of ADCC and especially ADCD for the beta and gamma VOCs.

6. I realize that the focus of this manuscript is on analysis of Fc-effector function and not neutralization, but I think it would be important to show that neutralization responses are indeed reduced across the variants (maybe at least comparing Wuhan to Omicron), which could further support the authors' conclusions that Fc-effector function remains intact across variants.

The revised manuscript now includes neutralization titers for a subset of vaccinated (n=23) and

convalescent(n=26) subjects against Wuhan and Omicron viruses (**new Figure 1E**, also below as **Response to Review Figure 2**). As anticipated, neutralization activity against the omicron variant was undetectable in nearly all subjects.

Response to Review Figure 2 and Revised Manuscript Figure 1E. Neutralization titers (NT₅₀) observed for vaccinated (left) and convalescent (right) subject serum samples against Wuhan (black) and Omicron (orange) strains. Limit of detection (LOD) is indicated in the dotted line. Red bar indicated median. Statistical significance was defined by Mann-Whitney test with (****p<0.0001).

This new data in the resubmission makes this very clear.

7. The data presented in Figure 5 demonstrates a strikingly elevated OC43-specific functional activity in convalescent individuals but not vaccinated – are antibodies against OC43 also elevated? The near complete loss of activity when assayed using the S2P suggests that the stabilized spike completely occludes the epitopes that may be recognized when presented as a monomer (is the S from Sino Biologics a monomer?) or just the S2 alone. Is this specific to OC43? I saw that the authors also include these different domains for SARS-CoV-2 (spike, S2P, the hexapropyl, S1 and S2) in their Fc array. Is there any difference in functional activity between these as well? Alternatively, could there be a specific crossreactive epitope that is only induced with SARS-CoV-2 infection that is occluded in the stabilized OC43 spike?

Yes, binding antibody responses to unstabilized forms of OC43, specifically the S2 domain, were also elevated in convalescent but not vaccinated subjects, as illustrated below (**Response to Review Figure 3, Supplemental Figure 3**). These results are consistent with more comprehensive prior reports (Crowley et al., eLife 2022, <https://doi.org/10.7554/eLife.75228>) investigating cross-reactivity. We also note that OC43 is not unique in this regard. For example, the other beta-CoV tested, HKU1, also showed evidence of elevated functional antibody responses (**Figure 5C, ADCP**) among convalescent but not vaccinated subjects.

The opening statement, “Yes, binding antibody responses to unstabilized forms of OC43, specifically the S2 domain, were also elevated in convalescent but not vaccinated subjects, as illustrated below,” seems to not fit with supplementary Figure 3. The IgG levels of the vaccinated cohort appear nearly equivalent to the convalescent cohort for OC43 S and S2 binding. For the IgG binding to OC43 S, the vaccinated cohort is even slightly greater than convalescent (mean log MFI). So, does the data indicate IgG binding to OC43 antigens (Supplementary Figure 3) is nearly equivalent between the vaccinated and convalescent cohorts, while for all effector functions (Figure 5C; ADCP, ADCC, ADCD) the convalescent cohort is superior?

For both the convalescent and vaccinated samples the reactivity with the stabilized S2P antigen is reduced, but there is no statistical analysis provided. There are no statistical analyses of supplemental Figure 1,2,3,4 to guide the reader to what are significant comparisons.

We agree that the differing induction of endemic CoV-cross-reactive antibodies is intriguing. Our studies are consistent with differing antigenicity among spike domains and conformational states. Subdomains and distinct spike conformations were not available for all CoV strains (only those described in **Supplemental Table 2**). We include below the IgG binding profiles of all tested forms of the Wuhan strain (**Response to Review Figure 4A, Supplemental Figure 4A**).

We have also tested effector functions across different Wuhan domains/conformations (**Response to Review Figure 4B**, now also included as **Supplemental Figure 4B**).

The authors have provided substantial additional data for this epitope reduction with the stabilized spike.

8. I’m a little surprised that there is minimal functional activity induced against HKU1 (for ADCC and ADCD) NL63 or 229E given the global presence of these viruses. Did the authors include a positive control for those particular antigens (a seropositive individual or IVIG, which should have some NL63 or 229E specific antibodies, for example) just to ensure that the effector assays are working well with these antigens?

We included either or both a high concentration of IVIG and a monoclonal Ab as positive controls, as presented below for endemic CoV (**Response to Review Figure 5 and Supplemental Figure 7**).

Functional responses were low, perhaps unsurprising given the considerably lower binding antibody

signal associated with these specificities as compared to SARS-CoV-2 antigens and the dilute serum used for analysis.

The authors have provided this additional data with an appropriate low dilution Ivlg control to validate activity in the functional assays.

9. The finding that IgA breadth is similar between infection and vaccination may have important implications for the role of different isotypes in mediating protection, especially in mucosal protection. The authors have not evaluated the potential role of IgA in mediating effector function across the variants. While THP-1 cells can express FcαR, so it is possible that IgA may be contributing to phagocytosis in these cells, perhaps neutrophil-mediated phagocytosis could be also explored for a subset of subjects (or binding to FcαR?)

We agree with the reviewer that IgA has an important role in the antibody response following vaccination and infection (Sterlin et al., Sci Trans Med, 2021, doi: 10.1126/scitranslmed.abd2223). Previous work from our group has explored ADNP activity (Butler et al., Front Immunol, 2021, doi: 10.3389/fimmu.2020.618685), which was observed to correlate with IgA and FcαR binding levels in serum. Given the number of antigen-specificities tested, we did not consider assays that required fresh primary effector cells. However, in the revised manuscript we now report FcαR binding (**Response to Review Figure 6 and Supplemental Figure 1**).

Response to Review Figure 6 and Supplemental Figure 1. FcαR responses to VOCs following vaccination or infection. Median fluorescent intensity of SARS-CoV-2 VOC spike- (left) and RBD- (right) specific antibody binding to FcαR as defined by multiplex assay. Responses among SARS-CoV-2 naive subjects are shown in black. Bar indicates median response.

FcαR binding activity for spike and RBD across the major VOC are now presented and functional significance has been shown in a previous publication (Butler et al., 2021).

Minor:

1. Figure 1A – the blue and green colors used in the figure may be difficult to parse for those with colorblindness or when viewed/printed in black and white. Perhaps the authors could use different shapes as well?

We have updated the UMAP plot with different shapes as suggested.

OK.

2. Table 1 – I didn't see the reference for Crowley et al. AI 2021 in the reference list, but I wanted to take a look at that paper to see if the group had previously reported the neutralization for this cohort.

Neutralization data for a subset of samples in this study are now included as **Figure 1E**.

OK.

3. Figure 2 – can the authors clarify whether the MFI on the X axis is the MFI of a single antigen or a composite of all antigens?

In **Figure 2**, the X axis is a composite, which is now clarified in the legend.

OK.

4. Figure 4 - The authors should include a line or indicate where the naive individual responses fall to allow the reader to understand whether the baseline responses against a given variant.

See **Response to Review Figures 3 and 4**. Additionally we have included **Response to Review Figure 7 and Supplemental Figure 2** to show naive functional response to SARS-CoV-2 variants.

OK.

5. Days post vaccine/days post PCR+/symptom onset – are these statistically equivalent?

These are not statistically equivalent. We include this result as **Response to Review Figure 8 and Supplemental Figure 5**.

Response to Review Figure 8 and Supplemental Figure 5. Days post second vaccine dose and post symptom onset. Days post second vaccine dose for vaccinated subjects (green) and days post SARS-CoV-

2 positive PCR result for convalescent subjects (blue). Statistical significance was defined by Mann-Whitney test with ($***p < 0.0001$).

This is unclear. Do the authors mean the difference IS statistically significant ($***p < 0.0001$)? Reviewer 1's question relates to how differences in the timing between infection and sampling, and between 2nd dose vaccination and sampling may contribute to the observed differences between the vaccinated and convalescent groups. The authors should comment.

6. Were the pregnant people vaccinated/infected prior to pregnancy or during pregnancy?

Pregnant individuals were vaccinated during the third trimester of pregnancy (lines 348-350).

OK.

Prior Reviewer 1 commentary are in black text.

Prior Author responses are in red text.

New review comments in blue text.

New Author responses in green text.

Reviewer 1 provides a highly favorable assessment of this paper. The authors have provided new data sets in response to many of Reviewer 1's questions. Only some small changes to the manuscript text are suggested. Some clarification of a couple of the author's responses could be helpful, but in my opinion does not require another round of review. Statistical analysis of the supplementary figures should be provided. Questions are in major and minor categories.

Major

1. The methods mention that 4 vaccinated individuals had evidence of prior infection. Were these individuals included in the analysis? As hybrid immunity induced by infection followed by vaccination boosts levels of antibodies, which has been observed by many groups, it may be best to remove those 4 individuals from the analysis.

We did include the four individuals with prior infection. We agree there have been numerous publications highlighting differences in hybrid immunity. In the context of our study, these subjects were maintained on the basis of lacking distinction from other vaccinated subjects (e.g. in the UMAP analysis).

Line 123f.

It is not made clear in the text that 4 individuals having hybrid immunity are included in the vaccine group. As reviewer 1 asks for clarification on this point it would be helpful for the text to make this clear and that their inclusion or exclusion did not affect the outcomes.

We thank the reviewer for raising this concern. To clarify we have added the following lines to the manuscript and included it here as well:

"These subjects were included in analysis as they were not clearly distinct from other vaccinated subjects in multivariate (UMAP) analysis." Line 325

2. The UMAP in Figure 1A shows a beautiful split between vaccinated and convalescent populations. The convalescent population is further split into two populations. What are the antibody features driving that split? Looking at Figure 1B, it looks as if there might be two populations in some of the antibody measures. Alternatively, did these people experience any differences in disease severity? The middle UMAP population looks to be comprised mostly of pregnant people. Did the authors also include parameters such as location, disease severity, days post-PCR+/days post 2nd dose? Were the effector function responses presented in Figure 4/5 also included in the UMAP?

We also found the split among convalescent subjects to be an interesting result of the UMAP analysis, which relied on IgM, IgA, and IgG antibody binding to spike and RBD SARS-CoV-2 strains. The reviewer is correct to suggest that the bimodal distribution in antibody binding to some of the variant antigens, particularly for RBD, appears to be a major driver, as described in line 141-145. Relationships between UMAP clusters and available clinical/demographic characteristics such as location, disease severity, age, pregnancy, time since infection, etc., were not apparent.

Can line 145 be extended to also indicate other associations were not apparent?

We have now included the following line in the text to clarify this point:

"With the exception of time since most recent antigen exposure, which differed between vaccinated and convalescent individuals, relationships between cluster group and other available clinical/demographic characteristics were not apparent." Line 114

3. The multiplexed approach to measure antigen-specific antibody levels has the advantage of measuring

multiple specificities in the same well. Are the authors concerned about competition of binding to variant antigens by the same antibodies, potentially leading to reduced responses against one antigen vs another? Do the antibodies have equivalent affinity across the variants, or is affinity skewed towards one variant over another?

Experimentally, testing in the context of extremely high degrees (eg: 50-plex) of highly redundant (eg: identical) antigens can reduce signal in these assays. Fortunately, while we cannot exclude an effect of signal suppression based on multiplexing, all samples were tested under the same conditions and thus subject to the same degree of competition.

As to insights into antibody affinity, unfortunately, signal in this assay relates to both antibody prevalence and affinity, leaving us unable to isolate affinity to support comparison across different antigen-specificities

Reviewer 1 has asked an interesting question on multiplexing methodology to which the authors have responded appropriately, but this should not affect the data presented or the text.

4. As IgM is a potent inducer of ADCD, were the authors surprised at the drop-off in ADCD across variants in the convalescents, especially since IgM levels against the variants and breadth score were elevated in convalescents? What does antibody binding to C1q look like or to MBL?

Elevated ADCD among convalescents is consistent with their higher levels of SARS-CoV-2-specific IgM. Both the drop-off in activity across variants, and against different conformations of Wuhan Spike (**Supplemental Figure 4**) are of interest. In our experience, the ADCD assay exhibits a threshold-like effect, and we hypothesize that the levels of antibody against VOC may fall in the very steep part of curve.

Antigen-specific antibody binding to MBL was not evaluated in this study; antigen-specific antibody binding to C1q is presented below (**Response to Review Figure 1, part of Supplemental Figure 1**).

C1q responses to VOCs following vaccination or infection. Median fluorescent intensity of SARS-CoV-2 VOC spike- (A) and RBD- (B) specific antibody binding to C1q as defined by multiplex assay. Responses among SARS-CoV-2 naïve subjects are shown in black. Bar indicates median response.

The authors have responded with a comprehensive data set of C1q binding for the cohorts which adds to an already comprehensive study. At present there is not my knowledge a way of dissecting IgG and IgM contribution to ADCD or C1q binding, although some specific inhibitors do exist- which might be used. The fall in ADCD in the convalescents is likely accounted for by responses being steep with respect to opsonizing concentration, but it would be helpful to explain this in the text for the reader.

We have elaborated on relationships between Ig levels and complement deposition profiles as follows:

“Among effector functions tested, complement deposition, which often exhibits a steep dose-response profile, was the activity most strongly impacted by strain differences, for example, showing high activity in convalescent subjects only for Wuhan and alpha strains. This function was also the most sensitive to serum concentration, and perhaps as a result, showed greater differences an activity levels across VOC.”
Line 189

Further to this aspect pointed out by reviewer 1, it is very interesting that the higher IgG1/3 binding to Wuhan spike/RBD by the vaccinated group over the convalescents is matched by higher C1q binding (Supplemental Figure 1), however this higher C1q binding does not result in higher ADCD, since the convalescent cohort has clearly higher ADCD than the vaccinated (Supplemental Figure 4). I naively expected trends in C1q binding to result in matching trends in ADCD. I think this is an aspect that other readers would not anticipate, and so the authors could consider if it is useful to point out in the text. I.e. vaccinated group has highest IgG binding to Wuhan and highest C1q binding, but convalescent ADCD is superior.

The finding of elevated ADCD for the convalescent subjects against the wild-type S antigen we agree was an interesting result. This can most likely be explain by the elevated levels of IgM antibodies seen in the convalescent subjects compared to vaccinated as IgM is a potent inducer of ADCD. We are

currently working with a cohort of convalescent maternal subjects with matched cord blood samples. This work will help us examine the role IgM plays in ADCD as the maternal samples will have high levels of IgM antibodies and the infant samples will have no IgM due to negligible transfer across the placenta.

We have added the following line in the text to highlight the point raised:

“Despite lower levels of IgG1, IgG3, and C1q binding of spike/RBD-specific antibodies, convalescent subjects exhibited greater complement deposition compared with the vaccinated cohort against Wuhan spike, an observation that may relate to contributions from IgM, which was elevated among convalescent individuals.” Line 199

5. Figure 4 shows that ADCC/ADCD is reduced against Beta and Gamma RBD, but then increased back in Delta and Omicron. Are there any mutations that are common only to those variants that could explain these data?

Beta and Gamma RBD sequences share two mutations (E484K and N501Y) while also both having a mutation at position 417 albeit for different amino acids. The delta variant does not share these mutations/sites with Beta and Gamma. Omicron, however, does possess the K417N mutation also present in Beta as well as the N501Y mutation present in Beta and Gamma. Omicron also bears a mutation at position 484 like Beta and Gamma variants, though for a different amino acid. The only mutation shared between Omicron and Delta is T478K which is not present in Beta or Gamma variants (<https://doi.org/10.1016/j.cell.2022.01.001>). We agree with the reviewer that this is an interesting observation within our study, but find that these mutational patterns do not suggest an obvious explanation.

This feature of the data is pointed to in the text of the results in line 217f. It is perhaps helpful to the reader to similarly extend line 292 in the discussion to re-point the reader to this interesting diminution of ADCC and especially ADCD for the beta and gamma VOCs.

We have updated the text to reflect this observation also provided here:

“...and among VOC, beta and gamma RBD variants showed a greater decrease in ADCC and complement deposition activity than more distant VOC, a pattern that was not easily explained by mutational loads or identities.” Line 275

6. I realize that the focus of this manuscript is on analysis of Fc-effector function and not neutralization, but I think it would be important to show that neutralization responses are indeed reduced across the variants (maybe at least comparing Wuhan to Omicron), which could further support the authors' conclusions that Fc-effector function remains intact across variants.

The revised manuscript now includes neutralization titers for a subset of vaccinated (n=23) and convalescent (n=26) subjects against Wuhan and Omicron viruses (**new Figure 1E**, also below as **Response to Review Figure 2**). As anticipated, neutralization activity against the omicron variant was undetectable in nearly all subjects.

Response to Review Figure 2 and Revised Manuscript Figure 1E. Neutralization titers (NT50) observed for vaccinated (left) and convalescent (right) subject serum samples against Wuhan (black) and Omicron (orange) strains. Limit of detection (LOD) is indicated in the dotted line. Red bar indicated median.

Statistical significance was defined by Mann-Whitney test with (***p<0.0001).

This new data in the resubmission makes this very clear.

7. The data presented in Figure 5 demonstrates a strikingly elevated OC43-specific functional activity in convalescent individuals but not vaccinated – are antibodies against OC43 also elevated? The near complete loss of activity when assayed using the S2P suggests that the stabilized spike completely occludes the epitopes that may be recognized when presented as a monomer (is the S from Sino Biologics a monomer?) or just the S2 alone. Is this specific to OC43? I saw that the authors also include

these different domains for SARS-CoV-2 (spike, S2P, the hexapro, S1 and S2) in their Fc array. Is there any difference in functional activity between these as well? Alternatively, could there be a specific cross reactive epitope that is only induced with SARS-CoV-2 infection that is occluded in the stabilized OC43 spike?

Yes, binding antibody responses to unstabilized forms of OC43, specifically the S2 domain, were also elevated in convalescent but not vaccinated subjects, as illustrated below (**Response to Review Figure 3, Supplemental Figure 3**). These results are consistent with more comprehensive prior reports (Crowley et al., eLife 2022, <https://doi.org/10.7554/eLife.75228>) investigating cross-reactivity. We also note that OC43 is not unique in this regard. For example, the other beta-CoV tested, HKU1, also showed evidence of elevated functional antibody responses (**Figure 5C, ADCP**) among convalescent but not vaccinated subjects.

The opening statement, “Yes, binding antibody responses to unstabilized forms of OC43, specifically the S2 domain, were also elevated in convalescent but not vaccinated subjects, as illustrated below,” seems to not fit with supplementary Figure 3. The IgG levels of the vaccinated cohort appear nearly equivalent to the convalescent cohort for OC43 S and S2 binding. For the IgG binding to OC43 S, the vaccinated cohort is even slightly greater than convalescent (mean log MFI). So, does the data indicate IgG binding to OC43 antigens (Supplementary Figure 3) is nearly equivalent between the vaccinated and convalescent cohorts, while for all effector functions (Figure 5C; ADCP, ADCC, ADCD) the convalescent cohort is superior?

We apologize for this poorly worded response describing the IgG binding data. We intended to convey that the levels of OC43-specific IgG were relatively higher (than CoV-2-specific IgG) among convalescent subjects. As noted above, we agree that similar levels of IgG binding to OC43 antigens in each conformation tested were observed. This distinction between binding and functional assay results is notable, and given differences in the isotypes of responses toward endemic CoV, is another point. The manuscript described this data correctly, but we have elaborated as follows:

*“Interestingly, the differences in antibody function were greater than might have been expected based on binding IgG antibody profiles observed across endemic CoV antigens (**Supplemental Figure 3**), perhaps again reflecting contributions from other isotypes.”* Line 220

For both the convalescent and vaccinated samples the reactivity with the stabilized S2P antigen is reduced, but there is no statistical analysis provided. There are no statistical analyses of supplemental Figure 1,2,3,4 to guide the reader to what are significant comparisons.

We have now included a supplemental table with all statistical analysis for each figure in the main text and supplement. Supplemental Data File 1 (raw data) and Supplemental Data File 2 (statistical comparisons) are now referred to in the Data Analysis and Statistical Quantification section. Line 426

We agree that the differing induction of endemic CoV-cross-reactive antibodies is intriguing. Our studies are consistent with differing antigenicity among spike domains and conformational states. Subdomains and distinct spike conformations were not available for all CoV strains (only those described in **Supplemental Table 2**). We include below the IgG binding profiles of all tested forms of the Wuhan strain (**Response to Review Figure 4A, Supplemental Figure 4A**).

We have also tested effector functions across different Wuhan domains/conformations (**Response to Review Figure 4B, now also included as Supplemental Figure 4B**).

The authors have provided substantial additional data for this epitope reduction with the stabilized spike.

8. I’m a little surprised that there is minimal functional activity induced against HKU1 (for ADCC and ADCD) NL63 or 229E given the global presence of these viruses. Did the authors include a positive control for those particular antigens (a seropositive individual or IVIG, which should have some NL63 or 229E specific antibodies, for example) just to ensure that the effector assays are working well with these antigens?

We included either or both a high concentration of IVIG and a monoclonal Ab as positive controls, as presented below for endemic CoV (**Response to Review Figure 5 and Supplemental Figure 7**).

Functional responses were low, perhaps unsurprising given the considerably lower binding antibody signal associated with these specificities as compared to SARS-CoV-2 antigens and the dilute serum used for analysis.

The authors have provided this additional data with an appropriate low dilution IVIG control to validate activity in the functional assays.

9. The finding that IgA breadth is similar between infection and vaccination may have important implications for the role of different isotypes in mediating protection, especially in mucosal protection. The authors have not evaluated the potential role of IgA in mediating effector function across the variants. While THP-1 cells can express FcαR, so it is possible that IgA may be contributing to phagocytosis in these cells, perhaps neutrophil-mediated phagocytosis could be also explored for a subset of subjects (or binding to FcαR?)

We agree with the reviewer that IgA has an important role in the antibody response following vaccination and infection (Sterlin et al., *Sci Trans Med*, 2021, doi: 10.1126/scitranslmed.abd2223). Previous work from our group has explored ADNP activity (Butler et al., *Front Immunol*, 2021, doi: 10.3389/fimmu.2020.618685), which was observed to correlate with IgA and FcαR binding levels in serum. Given the number of antigen-specificities tested, we did not consider assays that required fresh primary effector cells. However, in the revised manuscript we now report FcαR binding (**Response to Review Figure 6 and Supplemental Figure 1**).

Response to Review Figure 6 and Supplemental Figure 1. FcαR responses to VOCs following vaccination or infection. Median fluorescent intensity of SARS-CoV-2 VOC spike- (left) and RBD- (right) specific antibody binding to FcαR as defined by multiplex assay. Responses among SARS-CoV-2 naive subjects are shown in black. Bar indicates median response.

FcαR binding activity for spike and RBD across the major VOC are now presented and functional significance has been shown in a previous publication (Butler et al., 2021).

Minor:

1. Figure 1A – the blue and green colors used in the figure may be difficult to parse for those with colorblindness or when viewed/printed in black and white. Perhaps the authors could use different shapes as well?

We have updated the UMAP plot with different shapes as suggested.

OK.

2. Table 1 – I didn't see the reference for Crowley et al. AI 2021 in the reference list, but I wanted to take a look at that paper to see if the group had previously reported the neutralization for this cohort.

Neutralization data for a subset of samples in this study are now included as **Figure 1E**.

OK.

3. Figure 2 – can the authors clarify whether the MFI on the X axis is the MFI of a single antigen or a composite of all antigens?

In **Figure 2**, the X axis is a composite, which is now clarified in the legend.

OK.

4. Figure 4 - The authors should include a line or indicate where the naive individual responses fall to allow the reader to understand whether the baseline responses against a given variant.

See **Response to Review Figures 3 and 4**. Additionally we have included **Response to Review Figure 7 and Supplemental Figure 2** to show naive functional response to SARS-CoV-2 variants.

OK.

5. Days post vaccine/days post PCR+/symptom onset – are these statistically equivalent?

These are not statistically equivalent. We include this result as **Response to Review Figure 8 and Supplemental Figure 5**.

Response to Review Figure 8 and Supplemental Figure 5. Days post second vaccine dose and post symptom onset. Days post second vaccine dose for vaccinated subjects (green) and days post SARS-CoV-2 positive PCR result for convalescent subjects (blue). Statistical significance was defined by Mann-Whitney test with ($****p < 0.0001$).

This is unclear. Do the authors mean the difference is statistically significant ($****p < 0.0001$)? Reviewer 1's question relates to how differences in the timing between infection and sampling, and between 2nd dose vaccination and sampling may contribute to the observed differences between the vaccinated and convalescent groups. The authors should comment.

We apologize for the confusion. The days post second dose compared with time since infection are statistically significant, as noted at the opening of our response to point 5. We believe that inclusion of Supplemental Figure 5 makes this clear, and note this potential confounding factor as follows:

"While pregnant subjects completed their vaccination series in the third trimester, and most convalescent subjects reported symptoms or tested positive in their third trimester, elapsed time since most recent SARS-CoV-2 antigen exposure differed between cohorts (Supplemental Figure 5), which may impact some of the observations reported..." Line 332.

6. Were the pregnant people vaccinated/infected prior to pregnancy or during pregnancy?
Pregnant individuals were vaccinated during the third trimester of pregnancy (lines 348-350).

OK.

Reviewer #2 (Remarks to the Author):

The revised manuscript by Hederman et al. has addressed my questions raised to the original submission.

New data includes correlating ADCC and the Jurkat reporter, neutralisation of omicron and Fc α RI binding enhances the submission.

Altogether it is a comprehensive and important analysis showing superior antibody effector functions induced by vaccination over that from infection by SARS-CoV-2. The likely significance of these effector functions is highlighted by the ineffectual neutralization of recent VOCs.

Minor points,

line 156 "Figure 1F" should be 1E

We thank the reviewer for noticing and have fixed this mistake.

The symbols in the Y axis titles of supplementary Figure 1 are not rendered correctly.

We have fixed this issue and thank the reviewer for noticing.

REVIEWERS' COMMENTS

Reviewer #2 (Remarks to the Author):

The authors have responded appropriately to all the reviewers questions. Additions to the text on the last issues makes it clearer for the reader.